# Thermal state and evolving geodynamic regimes of the Meso- to Neoarchean North China Craton

Guozheng Sun[1], Shuwen Liu [1 ✉], Peter A. Cawood [2 ✉], Ming Tang[1], Jeroen van Hunen [3], Lei Gao[1], Yalu Hu[1] & Fangyang Hu[4,5,6]

Constraining thickness and geothermal gradient of Archean continental crust are crucial to understanding geodynamic regimes of the early Earth. Archean crust-sourced tonalitic–trondhjemitic–granodioritic gneisses are ideal lithologies for reconstructing the thermal state of early continental crust. Integrating experimental results with petrochemical data from the Eastern Block of the North China Craton allows us to establish temporal–spatial variations in thickness, geothermal gradient and basal heat flow across the block, which we relate to cooling mantle potential temperature and resultant changing geodynamic regimes from vertical tectonics in the late Mesoarchean (~2.9 Ga) to plate tectonics with hot subduction in the early to late Neoarchean (~2.7–2.5 Ga). Here, we show the transition to a plate tectonic regime plays an important role in the rapid cooling of the mantle, and thickening and strengthening of the lithosphere, which in turn prompted stabilization of the cratonic lithosphere at the end of the Archean.

[1] Key Laboratory of Orogenic Belts and Crustal Evolution, Ministry of Education, School of Earth and Space Sciences, Peking University, Beijing, PR China. [2] School of Earth, Atmosphere and Environment, Monash University, Melbourne, VIC, Australia. [3] Department of Earth Sciences, Durham University, Durham, UK. [4] Key Laboratory of Mineral Resources, Institute of Geology and Geophysics, Chinese Academy of Sciences, Beijing, China. [5] Innovation Academy for Earth Science, Chinese Academy of Sciences, Beijing, China. [6] Department of Geosciences, University of Arizona, Tucson, AZ, USA. ✉email: swliu@pku.edu.cn; peter.cawood@monash.edu

The Archean Earth (4.0–2.5 Ga) was characterized by higher upper-mantle potential temperatures ($T_p$), higher mantle heat flow, and significantly less differentiated continental lithosphere than the present day[1–3]. Numerical models using these inferred Archean conditions have suggested a geodynamical evolution from "no-subduction" to "pre-subduction" and then to "modern subduction" regimes with the progressive decrease of the mantle potential temperature[4–6]. However, the timing and conditions under which the Earth transitioned to a plate tectonic regime are controversial due to, and dependent on, poorly constrained estimates for the secular cooling of Earth's mantle[7–13]. Crustal thickness, Moho temperature, and heat flow are direct reflections of the thermal conditions of Earth's lithosphere and profoundly influence lithospheric rheology and tectonics. Therefore, constraining these parameters will provide important new insight in Archean geodynamic regimes[14–17].

Tonalite–trondhjemite–granodiorite (TTG) gneisses constitute a dominant part of all the granite-greenstone belts and high-grade terranes in globally preserved Archean cratons. Therefore, they play a crucial role in understanding the formation and evolution of ancient continental crust[18–20]. Recent investigations suggest that Archean TTGs may be derived from (1) fractional crystallization of mafic melts from an enriched lithospheric mantle[21–23]; (2) partial melting of hydrated mafic rocks at the base of continental crust (Fig. 1a)[24,25]; and, (3) melting of subducted oceanic slabs (including steep and shallow subduction) (Fig. 1b, c)[26], with shallow subducted oceanic slabs underplating along the crust-mantle boundary and eventually incorporated into the continental crust. It is worth noting that $H_2O$-fluxed melting plays a crucial role in the formation of slab-derived TTG melts during steep subduction[26], which contributes to the compositional diversity of the TTGs. Moyen[19] classified the Archean (~3.5–2.5 Ga) TTGs into high-pressure (HP), medium-pressure (MP) and low-pressure (LP) groups based on geochemical indicators (i.e., Sr contents, Sr/Y, and La/Yb ratios). Most TTGs (~80%) belong to the LP and MP groups that originated from the thickened lower crust (around 30–45 km depth), whereas the remaining 20% form HP TTGs and were interpreted to be derived from subducting slabs (>60 km)[19]. This pressure classification scheme for TTGs has wide applicability[27], but its robustness has been questioned. For instance, Johnson et al.[25] demonstrated that the amount of garnet in residues is strongly dependent on the Mg# values (Mg# = Mg/(Mg + Total $Fe^{2+}$) × 100) of the metabasaltic source rocks, and proposed that MP TTGs may be stable at 0.7 GPa (~25 km). However, these Pilbara metabasaltic source rocks have high TFeO contents resulting in their Mg# values being significantly lower than the average value of TTGs, which expands the garnet stability field to lower pressure (<1.0 GPa)[28]. Furthermore, Smithies et al.[22] suggested that HP TTGs were not derived from partial melting of crustal materials, but from fractional crystallization of mafic melts that originated from the metasomatically enriched lithospheric mantle. These observations emphasize the importance of protolith composition and source water content, in addition to pressure and temperature, in generating TTGs.

The source rock composition, petrogenetic process, and tectonic setting of TTGs are extremely complicated, and only those TTG melts derived from the base of the crust are suitable for reconstructing the thermal state of the early Earth's continental crust. Studies in experimental petrology indicate that the TTGs derived from the melting of mafic crustal rocks are mainly formed at the root of the continental crust, which is close to the boundary between the crust and the lithospheric mantle (i.e., the Moho)[29,30]. In this way, the pressure (P) conditions of crustal-derived TTG melts (including the inferred shallow subduction slab-derived melts) may be used to estimate the minimum crustal thickness, and the temperature (T) conditions representing the

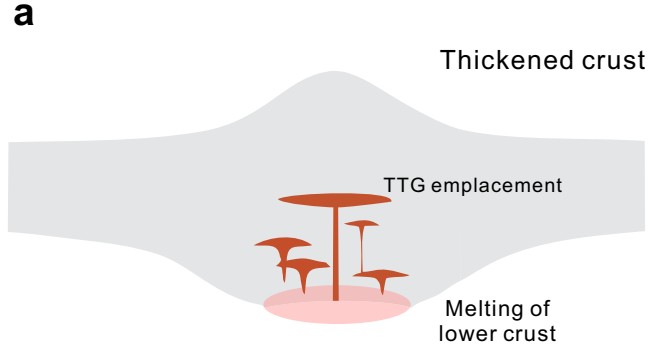

**a**

Thickened crust

TTG emplacement

Melting of lower crust

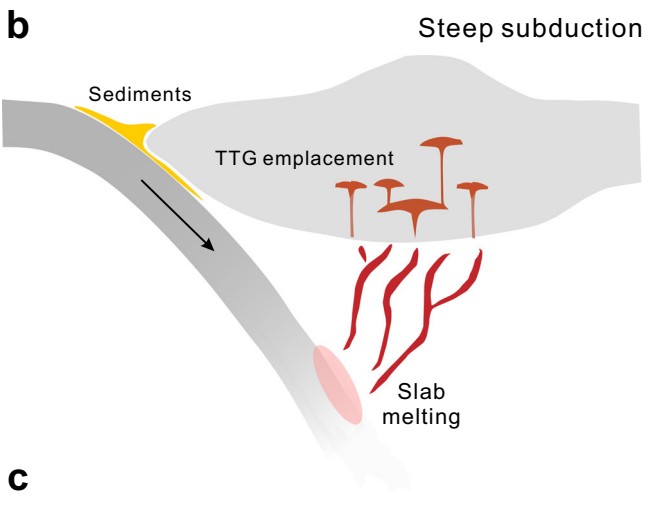

**b**

Steep subduction

Sediments

TTG emplacement

Slab melting

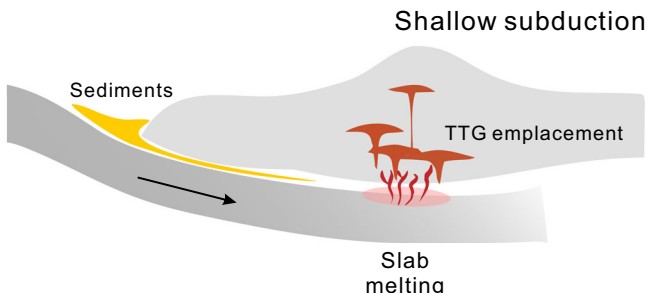

**c**

Shallow subduction

Sediments

TTG emplacement

Slab melting

**Fig. 1 Cartoon (not to scale) showing three different geodynamic settings for Archean TTG magma genesis. a** Melting of mafic rocks at the base of oceanic plateau/thickened crust; Slab melting at a convergent plate boundary at (**b**) steep and (**c**) shallow dip angles (after Palin et al.[26]).

lower limit of the Moho temperature. Following the geothermal models proposed by Chapman[31], we quantitatively calculate the basal heat flow ($q_B$) and Moho geothermal gradient assuming a steady-state conductive geotherm[32]. Therefore, the Archean crust-sourced TTGs are ideal lithologies for reconstructing the thermal state of early continental crust.

In this work, we compile the geochemical data for multiple episodes of Meso- to Neoarchean (~2.9, ~2.7, and ~2.5 Ga) TTGs from the Eastern Block of the North China Craton, and quantify the P–T conditions of the crustal-derived TTG melts by combining thermodynamic and trace element modeling. This allows the crustal thickness, Moho geothermal gradient, and basal heat flow ($q_B$) in the studied periods of the Archean to be calculated.

On the basis of these newly obtained crucial parameters, we propose a systematic evolution of Archean geodynamic regimes to explain the formation and evolution of the early continental lithosphere.

## Results

**Data selection.** This study is based on a dataset of 397 analyses (previously published and our latest data) of ca. 2.9 to 2.5 Ga TTGs from the Eastern Block of the North China Craton. These TTG samples were collected from Anhui, Zhongtiao, Dengfeng-Taihua, East Hebei-West Liaoning, South Jilin-North Liaoning, Zanhuang, Fuping, West Shandong, and Jiaodong terranes in the Eastern Block (Supplementary Fig. 1). For a detailed description of each terrane, see Supplementary Figs. 2–8.

We define TTGs as silica-rich granitoid ($SiO_2 > 64$ wt%) with high $Na_2O$ (3.0 wt% $\leq Na_2O \leq$ 7.0 wt%) and $Al_2O_3$ ($Al_2O_3 > 13$ wt%) contents[24]. A total of 287 TTG samples that align with this definition form a data subset of the TTGs. The selected samples plot into the tonalite, trondhjemite, and granodiorite fields in the An–Ab–Or diagram (Fig. 2a)[33]. In order to investigate the thermal state of continental crust, we selected from the data subset of TTGs those samples that are inferred to have formed by partial melting of lower crustal mafic rocks, based on the following criteria. Firstly, we removed samples with heterogeneous whole-rock Nd and zircon Hf isotopic compositions, which may be derived from magma mixing or from contamination of felsic crust as well as mantle peridotite during TTG magmatism. Secondly, we applied petrogenetic discrimination using the highly ($C^H$) and moderately ($C^M$) incompatible element ratio ($C^H/C^M$) versus ($C^H$) diagram. Combining with the isotopic features above, most TTG samples distribute along a straight line that suggests a partial melting formation process. Samples showing a horizontal trend or a discrete distribution have been excluded because they may experience either magma mixing or assimilation and fractional crystallization processes (Fig. 2b)[34]. Finally, the TTG samples derived from partial melting can be further divided into two categories according to the distribution characteristic of Mg# values (Fig. 2c)[29,35,36]. Low-magnesium TTGs exhibit relatively low MgO contents and Mg# values, and plot in the compositional range of experimentally obtained partial melts of amphibolite and eclogite. By contrast, high-magnesium TTGs display higher MgO contents and Mg# values than crustal partial melts obtained from the experiment, which, together with their high transition element (V, Cr, Ni, and Co) concentrations, indicate their interactions with peridotitic mantle (Fig. 2c; Supplementary Data 1). Therefore, the low-magnesium TTGs most likely originated from partial melting of basaltic rocks at the bottom of thickened crust, and their formation $P–T$ conditions may record the thermal states of early continental crust.

**Protolith characteristics.** Geochemistry and experimental petrology reveal that moderately enriched Archean tholeiites are ideal protoliths for TTGs[37]. The zircon Hf isotopic compositions of the dated TTG samples plot mainly between the CHUR (chondrite uniform reservoir) and depleted mantle lines and fall within the same crustal evolution range (Fig. 2d; Supplementary Data 2). These relationships imply that TTGs with different ages were likely derived from juvenile crustal sources with similar chemical compositions, namely, the lower crust of the North China Craton. In the $Al_2O_3/(FeO_{tot} + MgO) - 3 \times CaO - 5 \times (K_2O/Na_2O)$ source discriminate diagrams, almost all the TTGs plot into the field of melts derived from low-K mafic rocks (Fig. 2e)[38]. This further confirms that the Archean metamorphic low-K mafic rocks may be the most appropriate source rocks for these crust-derived TTG melts.

**Estimation of the thermal state for North China Craton.** Thermodynamic and trace element modeling was performed using the average compositions of ~2.9, ~2.7, and ~2.5 Ga low-K tholeiitic rocks with various water contents (1.2–2.0 wt%, defined by the average loss on ignition) from the Eastern Block of the North China Craton to better determine the $P–T$ conditions of the primary magma (Fig. 3). The mineral proportion of residual phases and melting degree were obtained under specific $P–T$ conditions by thermodynamic calculations. Then, we simulated multiple trace element compositions of the partial melts based on the mineral proportions in the residual phases, melting degree, and partition coefficient of trace elements using a simple batch partial melting model[39], up to the chemical composition of modeled melts close to those of actual TTG samples. Thus, we were able to quantitatively calculate the $P–T$ range of TTG melts. The standard uncertainty on $P–T$ estimation was cited to be ±1 kbar and ±50 °C at the 2 sigma level[40]. Assuming that the tectonic overpressure for the continental crust is negligible and that 1 GPa ≈33 km crustal depth, the pressure ($P$) range of crustal-derived TTG melts provides a minimum estimate of crustal thickness at that time. There are some variations in the estimation of a crustal thickness (±3.3 km, $2\sigma$) due to the intrinsic errors in the pressure estimation. As a result of this analysis, the paleo-crustal thickness was estimated to have evolved from 27 to 39 km in the late Mesoarchean to 33–62 km in the early Neoarchean and 33–59 km thick crust at the end of the Neoarchean (Table 1). Based on the newly obtained crustal thicknesses, and the temperature at the bottom of continental crust, we reconstruct the Archean continental geotherm assuming a steady-state conductive geotherm (Fig. 4)[31,32]. This yields basal heat flows and Moho geothermal gradients of 46–80 $mW/m^2$ and 18–31 °C/km for ~2.9 Ga TTG melts, 18–63 $mW/m^2$ and 7–24 °C/km for ~2.7 Ga TTG melts, and 20–50 $mW/m^2$ and 8–22 °C/km for ~2.5 Ga TTG melts (Table 1).

## Discussion

The trace element and thermodynamic modeling results reveal that the crustal thickness has changed significantly through time in the Eastern Block of the North China Craton (Fig. 5a). Crustal thickness increased rapidly from ~2.9 to ~2.7 Ga, then remained mostly constant during ~2.7 to ~2.5 Ga, with the exception of the North Liaoning-South Jilin, Jiaodong and West Shandong terranes, where the crustal thickness decreased about 10 km (Fig. 5a). The Moho temperature gradually decreased from ~850° to ~780 °C from ~2.9 to ~2.5 Ga (Fig. 5b). Archean continental geotherm modeling suggests that the Moho geothermal gradient and basal heat flow value decreased dramatically from ~2.9 to ~2.7 Ga and then remain unchanged from ~2.7 to ~2.5 Ga (Figs. 4 and 5c). Contour maps of crustal thickness and Moho geothermal gradients of the Eastern Block show that in the early Neoarchean (~2.7 Ga), the North Liaoning-South Jilin and West Shandong regions have thick crust but low Moho geothermal gradients, whereas the Jiaodong, Fuping, Zanhuang, and Dengfeng-Taihua terranes exhibit relatively thin crust but high Moho geothermal gradients (Fig. 6a, c). By the late Neoarchean (~2.6–2.5 Ga), the crustal thickness and Moho geothermal gradient display obvious spatial zonation; for instance, the western margin of the Eastern Block (East Hebei-West Liaoning, Dengfeng-Taihua, Zhongtiao) shows relatively thickened crust and low Moho geothermal gradients, but the eastern margin of the Eastern Block (North Liaoning-South Jilin, Jiaodong, West Shandong) exhibits relatively thin crust and high Moho geothermal gradients (Fig. 6b, d).

The operation, scale, and style of tectonic regimes in the formation and evolution of the Archean continental crust are mainly

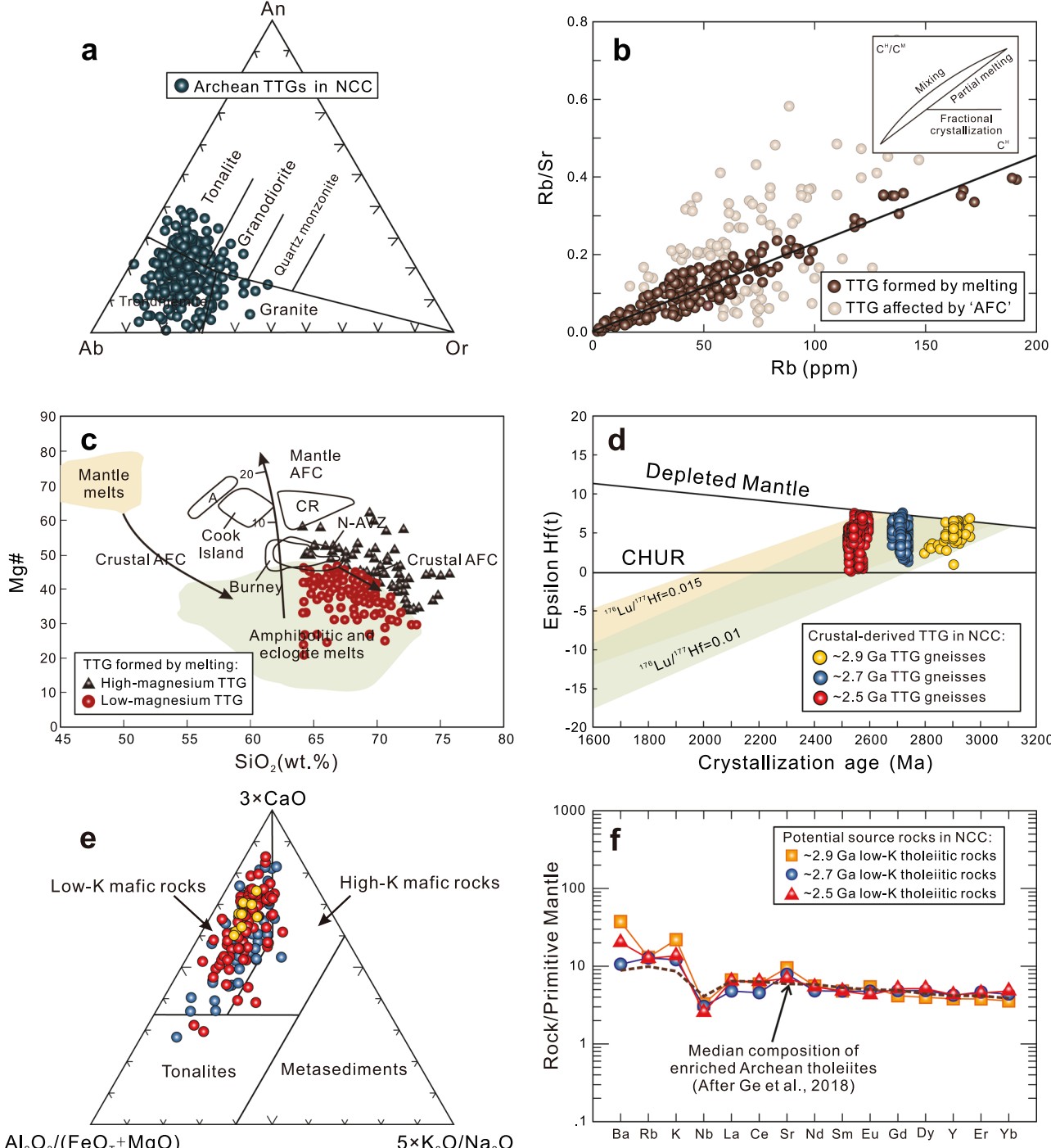

**Fig. 2 Geochemical diagrams for the Meso- to Neoarchean TTG gneisses from the Eastern Block of the North China Craton. a** An-Ab-Or diagram (after Barker[33]); **b** Rb/Sr versus Rb (ppm) diagram, the inset is a schematic $C^H/C^M$ versus $C^H$ plot, with $C^H$ and $C^M$ as concentrations of highly and moderately incompatible elements, respectively (after Schiano et al.[34]); **c** Mg# versus $SiO_2$ (wt.%) diagram (modified after Stern and Killian[36]). This also shows the field for pure crustal partial melts obtained in experimental studies by dehydration melting of amphibolitic rocks and eclogites[29,35]; **d** Plots of the zircon εHf(t) values versus crystallization ages (Ma) for the dated crustal-derived TTG samples (Supplementary Data 2); **e** Ternary diagram of $Al_2O_3/(FeO_T + MgO)$, $3 \times CaO$, and $5 \times (K_2O/Na_2O)$. The fields represent the composition of melts derived from a range of potential sources of tonalites, metasediments, and low- and high-K mafic rocks[38]; **f** Trace element patterns for the average composition of ~2.9 Ga, ~2.7 Ga and ~2.5 Ga low-K tholeiitic rocks from the Eastern Block of the North China Craton. The median values for the enriched Archean tholeiites are from Ge et al.[27].

dependent on the mantle temperature as shown by thermo-mechanical numerical experiments[4–6]. Nevertheless, such models do not constrain a timeframe for such elevated mantle temperatures. This study enables us to link those temperatures to

geological time. In the Archean continental geotherm model, the variation of basal heat flow ($q_B$) is mostly attributed to the mantle potential temperatures and the depth of the lithosphere-asthenosphere boundary (i.e., the lithospheric thickness) (Fig. 4).

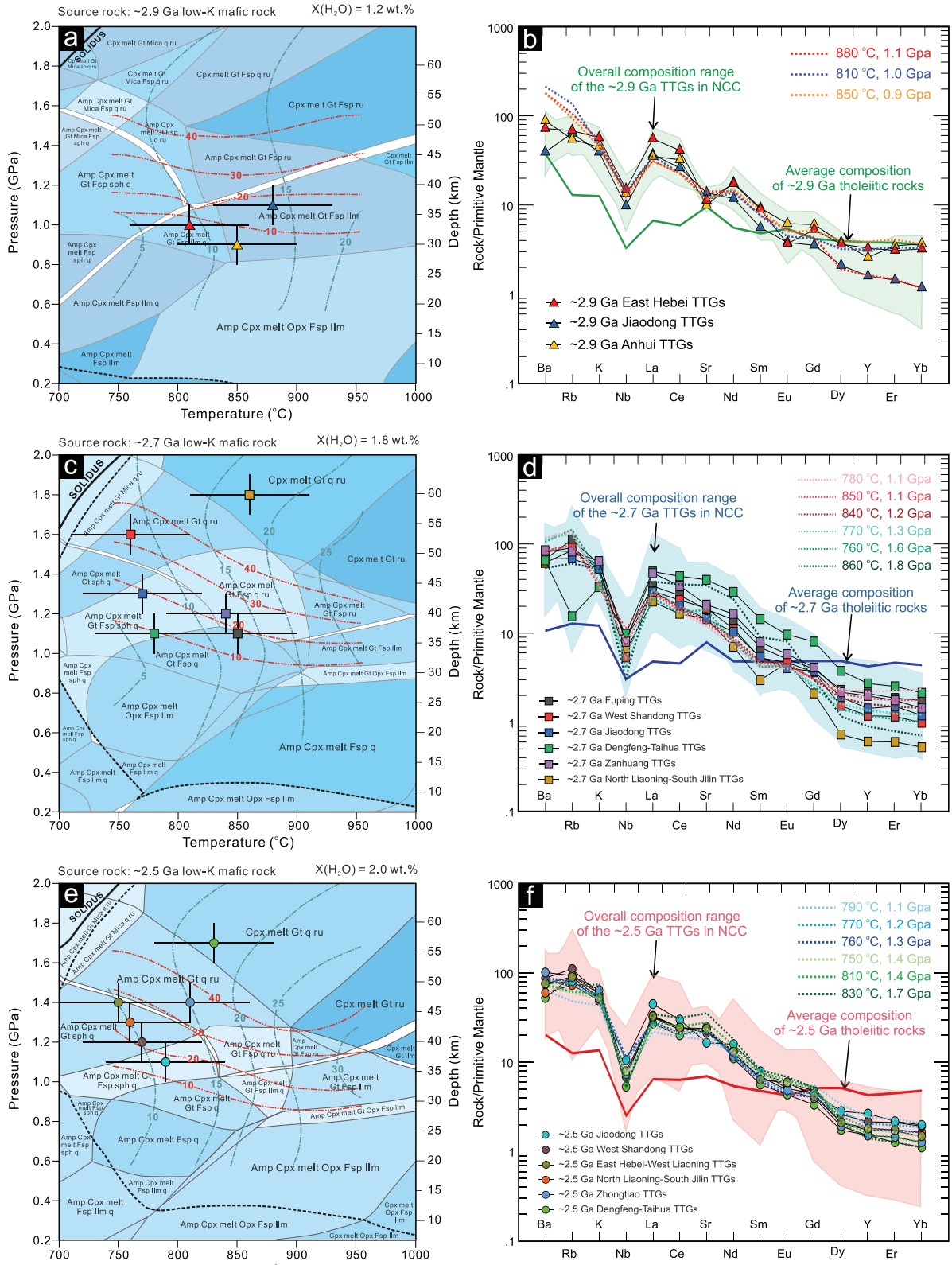

**Fig. 3 Thermodynamic and trace element modelling. a, c, e** Simplified *P–T* phase diagram for the average composition of ~2.9 Ga, ~2.7 Ga and ~2.5 Ga low-K tholeiitic rocks from the Eastern Block of the North China Craton, calculated with water contents of 1.2, 1.8 and 2.0 wt.%, respectively, corresponding to dehydration melting of metamorphosed mafic rocks. Green and red dashed lines show calculated degree of melting (wt.% of melt) and garnet proportion (%) in residue, respectively, black dotted lines mark water saturation of the system; **b, d, f** Trace element patterns of TTG melts calculated at different *P–T*–X(H₂O) conditions for the average composition of Meso- to Neoarchean low-K tholeiitic rocks. The pale green, blue and pink shaded areas indicate the overall compositional range of the ~2.9 Ga, ~2.7 Ga and ~2.5 Ga TTG melts, respectively.

**Table 1 Key values of calculated Meso- to Neoarchean thermal state of the Eastern Block, North China Craton.**

| Era and location | Melting temperature (°C) | Melting pressure (GPa) | Crustal thickness (km) | $q_B$ (mW m$^{-2}$) | Moho geothermal gradient (°C/km) |
|---|---|---|---|---|---|
| *~2.9 Ga* | | | | | |
| Jiaodong | 880 ± 50 | 1.1 ± 0.1 | 33–39 | 46–65 | 18–25 |
| Anhui | 850 ± 50 | 0.9 ± 0.1 | 27–33 | 55–80 | 21–31 |
| East Hebei | 810 ± 50 | 1.0 ± 0.1 | 30–36 | 46–67 | 18–26 |
| *~2.7 Ga* | | | | | |
| North Liaoning-South Jilin | 860 ± 50 | 1.8 ± 0.1 | 56–62 | 18–28 | 7–11 |
| West Shandong | 760 ± 50 | 1.6 ± 0.1 | 50–56 | 19–30 | 7–11 |
| Jiaodong | 770 ± 50 | 1.3 ± 0.1 | 40–46 | 29–43 | 11–17 |
| Fuping | 850 ± 50 | 1.1 ± 0.1 | 33–40 | 42–63 | 16–24 |
| Zanhuang | 840 ± 50 | 1.2 ± 0.1 | 37–43 | 37–53 | 14–20 |
| Dengfeng-Taihua | 780 ± 50 | 1.1 ± 0.1 | 33–40 | 37–57 | 14–22 |
| *~2.5 Ga* | | | | | |
| North Liaoning-South Jilin | 760 ± 50 | 1.3 ± 0.1 | 40–46 | 29–43 | 11–16 |
| West Shandong | 770 ± 50 | 1.2 ± 0.1 | 37–43 | 33–48 | 13–19 |
| Jiaodong | 790 ± 50 | 1.1 ± 0.1 | 33–40 | 38–50 | 15–22 |
| East Hebei-West Liaoning | 750 ± 50 | 1.4 ± 0.1 | 43–50 | 24–38 | 9–14 |
| Dengfeng-Taihua | 830 ± 50 | 1.7 ± 0.1 | 53–59 | 20–30 | 8–12 |
| Zhongtiao | 810 ± 50 | 1.4 ± 0.1 | 43–50 | 27–41 | 10–16 |

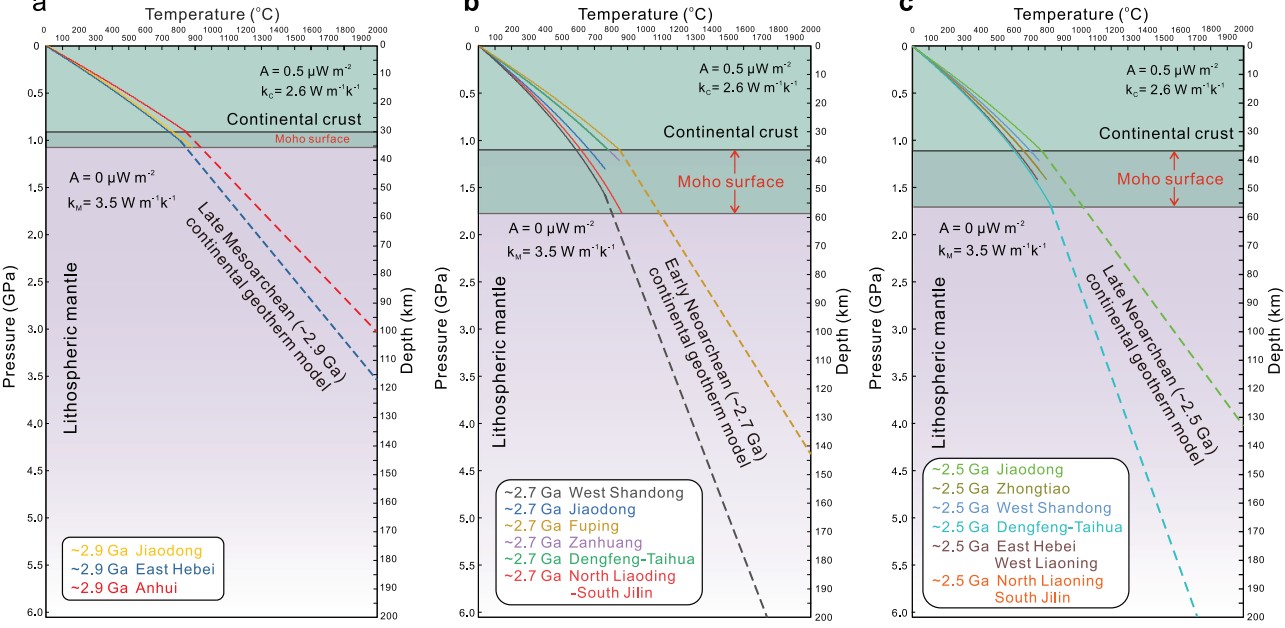

**Fig. 4 Generic model for the Meso- to Neoarchean thermal evolution of the North China Craton. a** Late Mesoarchean (~2.9 Ga) continental geothermal model; **b** early Neoarchean (~2.7 Ga) continental geothermal model; **c** late Neoarchean (~2.5 Ga) continental geothermal model. In these models, the Archean continental geotherm was constrained assuming a steady-state conductive geotherm, without considering the effects of heat convection of the asthenosphere mantle. The depth of Moho surface is determined by melting pressure of crustal-derived TTG melts, whereas the Archean lithospheric thickness is uncertain.

Keller and Schoene[41] proposed that the above two variables have a direct causal relationship in the lithospheric evolution process prior to 2.5 Ga. A decrease in mantle potential temperature may be linked to the emergence of thick lithosphere in the Archean. Combining our constraints on the Meso- to Neoarchean crustal thicknesses and thermal states with previous numerical experiments, petrological and structural studies, systematic three-stage evolution of the Archean geodynamic regime has been revealed in the Eastern Block of the North China Craton.

(1) The Eastern Block exhibits a relative thin crustal thickness of 27–39 km, high Moho geotherm of 18–31 °C/km, and high basal heat flow values of $q_B = 46$–80 mW/m$^2$ during ~2.9–2.8 Ga (Fig. 5a, c; Table 1), reflecting the higher mantle potential temperature and the thinner lithosphere thickness relative to younger time periods (Figs. 4 and 5). Three lines of evidence strongly support that a vertical tectonic regime played a dominant role in the late Mesoarchean: (a) the late Mesoarchean komatiites in west Shandong exhibit Al-depleted geochemical compositions,

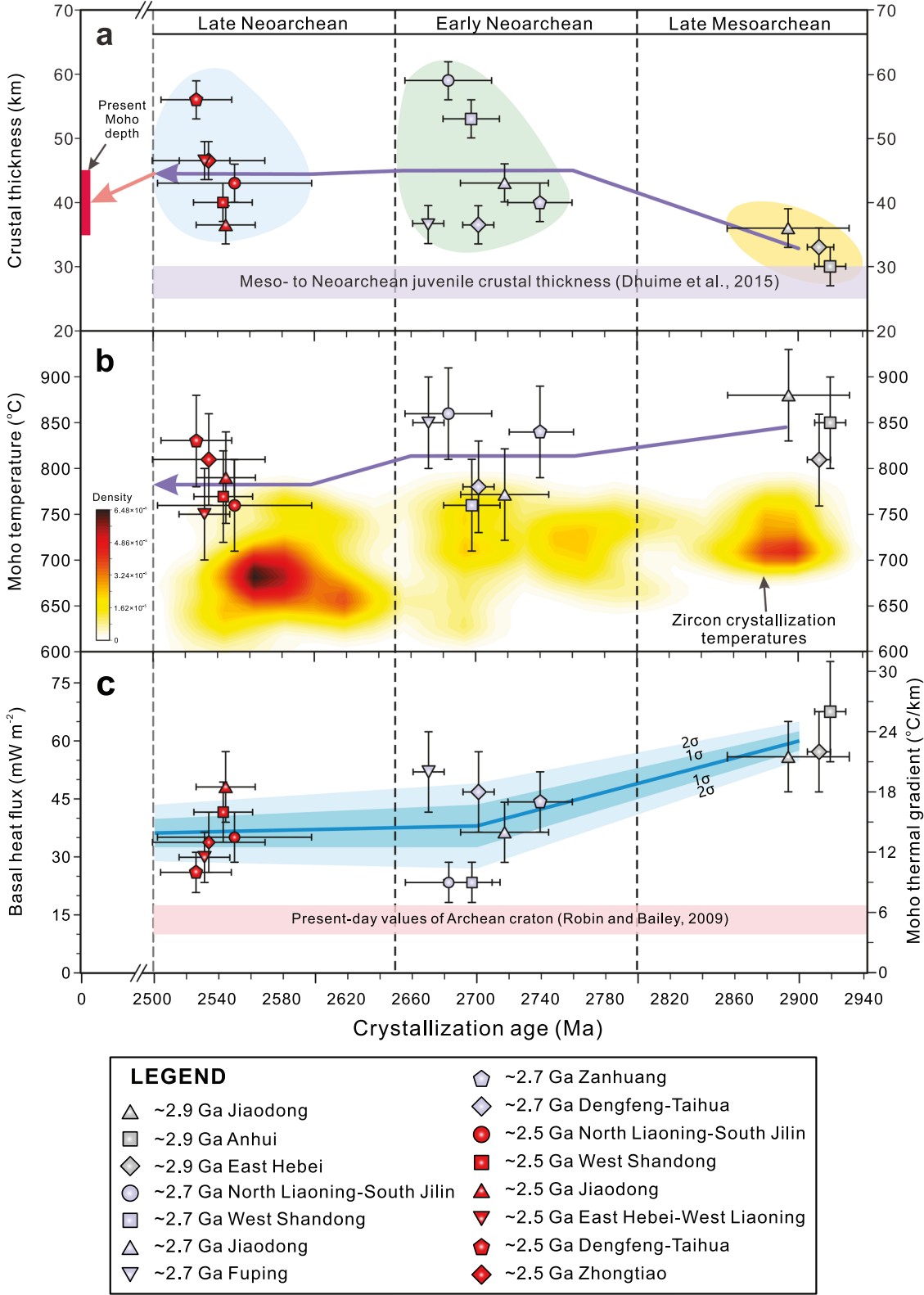

**Fig. 5 Changes of calculated crustal thickness, Moho temperature, basal heat flux and Moho geothermal gradient during ~2.9 Ga to ~2.5 Ga for the Eastern Block of the North China Craton. a** The present Moho depth is from McLennan[64], and the Meso- to Neoarchean juvenile crustal thickness is from Dhuime et al.[65]. **b** Zircon crystallization temperatures were calculated by Titanium-in-zircon geothermometer proposed by Ferry and Watson[66]. **c** The present-day basal heat flux values of Archean craton are from Robin and Bailey[67].

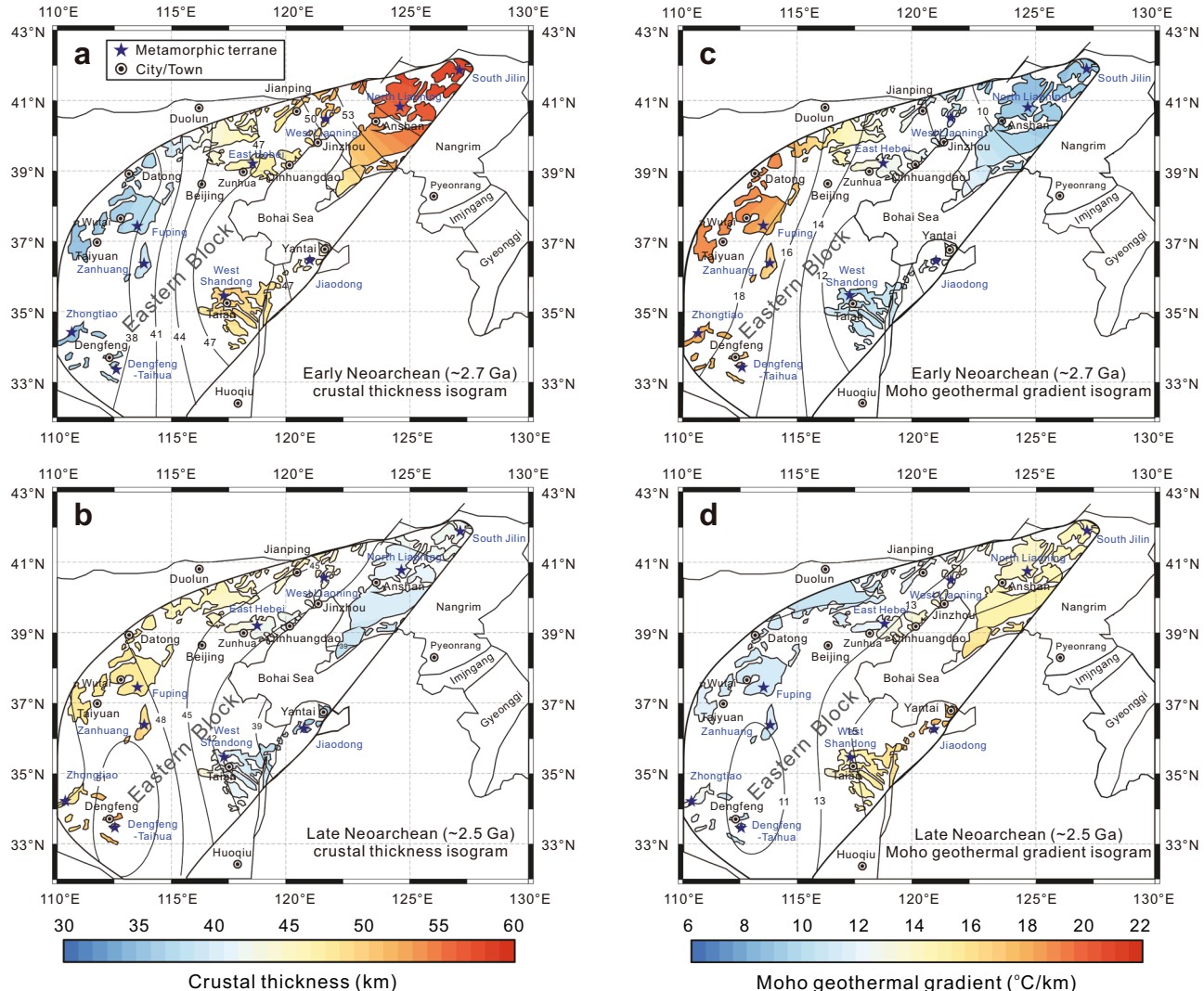

**Fig. 6 Contour color plots of crustal thickness and Moho geothermal gradient in the Eastern Block of the North China Craton at different periods (modified from Wang et al.[48]). a** Early Neoarchean (~2.7 Ga) crustal thickness isogram; **b** late Neoarchean (~2.5 Ga) crustal thickness isogram; **c** early Neoarchean (~2.7 Ga) Moho geothermal gradient isogram; **d** late Neoarchean (~2.5 Ga) Moho geothermal gradient isogram.

which could form by high-degree partial melting of the mantle[42]; (b) the granite-greenstone belt in the Eastern Block experienced ~2.85–2.80 Ga extensional deformation, which may be caused by a mantle upwelling[43]; and (c) high-mantle potential temperatures ($T_P$) in the late Mesoarchean has been suggested to favor cratonic formation by mantle upwelling[2,32].

(2) Crustal thickness increased rapidly from the late Mesoarchean (~2.8 Ga) and reached its maximum thickness of ~33–62 km by the early Neoarchean (~2.7 Ga), while the Moho geothermal gradient decreased from ~18–31 to ~7–24 °C/km (Fig. 5a, c; Table 1). During this crustal thickening period, the basal heat flow decreases significantly, reflecting rapid cooling of the convecting mantle and/or thickening of the lithosphere (Figs. 4 and 5). Mantle cooling would also promote lithosphere strengthening. This transition from the Mesoarchean into the Neoarchean is also characterized by a change from komatiite to calc-alkaline volcanic rock, and a switch in structural style from extension to compression, and has been related to plate tectonics being the dominant geodynamic mechanism[43]. However, the style of Archean plate tectonics (for example, subduction rate and subducted angle of the oceanic slab) in North China may have been

different from that of modern-day cold plate tectonics due to its slightly higher mantle temperature on the basis of the thermomechanical numerical modeling results and is often referred to as "hot subduction"[44,45]. Therefore, the Mesoarchean–Neoarchean transition (~2.8 Ga) is considered as a critical period for the transformation of the crust–mantle dynamics, in which a hot subduction regime (lateral plate movement) began to play an important role in the crustal growth and evolution in the Eastern Block of the North China Craton.

(3) A crustal stabilization period from ~2.7 to 2.5 Ga is marked by the crustal thickness of 33–59 km, Moho geothermal gradient of 8–22 °C/km, and basal heat flow of 20–50 mW/m² (Fig. 5a, c; Table 1). These values have decreased slightly from those of the preceding period and are considered to represent a likely continuation of the tectonic regime dominated by hot subduction (Figs. 4 and 5). In the thermomechanical numerical models of plate tectonics on the early Earth, as a subduction zone matures, mantle potential temperature continues to decrease whereas the strength of lithospheric plates gradually increases, resulting in the formation of larger-scale tectonic belts that resemble those produced by modern plate subduction[4–6]. By the

early to late Neoarchean, several lines of evidence indicate that the Eastern Block of the North China Craton underwent an orogenic cycle of subduction, collision, and delamination events, including: (a) the thermal state of continental crust is spatially zoned (Fig. 6b, d; Table 1), which is similar to the thermal structure of present-day continental subduction zones such as North Cordilleran and Central Andean[46]; (b) marked spatial zoning of metamorphic volcanic rocks, TTGs and K-rich granitoids occur along the northwestern side of the Eastern Block[47–49]; (c) Neoarchean paired metamorphic belts along the southern margin of the Eastern Block of the North China Craton are inferred to record subduction to collisional orogenesis at a convergent plate margin[50]; and, (d) in situ zircon O isotopes for Neoarchean metavolcanic rocks from the Dengfeng Complex indicates that altered oceanic crust was involved in the mantle source region, requiring recycling of oceanic crust into the mantle, presumably at a subduction zone[51].

At the end of Archean (~2.5 Ga), a large amount of K-rich granitoids were emplaced into the North China Craton along with a corresponding reduction in the proportion of TTGs[52]. Post-Archean high-Mg-K granitoids throughout the Eastern Block show similar geochemical signatures to Archean sanukitoids, and their magmas are considered to have been derived from mantle metasomatism by fluids as well as melts from subducting slab and sediments in late-orogenic settings[49,53,54]. These geological observations and crust-mantle interactions, together with relatively low mantle potential temperature and thick lithosphere, imply the development of rigid lithosphere and the final cratonization of the Eastern Block at the end of the Archean. Similar features have been noted in other cratons for the same time period, corresponding with their stabilization and inferred to mark a global transition to a predominantly plate tectonic regime[11,55].

## Methods

**Whole-rock geochemical and zircon U–Pb-Lu–Hf isotopic analysis.** Whole-rock major and trace elements were analyzed at the Key Laboratory of Orogenic Belts and Crustal Evolution, Ministry of Education, Peking University. Major oxides were determined using an automatic XRF spectrometer. Trace elements, including rare earth elements, were measured by ICP–MS with an Agilent 7500 instrument. Standards GSR-9 (diorite) and GSR-14 (granitoid gneiss) were used for analytical control. The analytical precisions for the major oxides and trace elements are 0.5 and 5%, respectively. Zircon U–Pb isotope dating and in situ trace element analyses were carried out on an Agilent 7500 quadruple-based ICP–MS with a GeoLas 193 nm laser at the Key Laboratory of Orogenic Belts and Crustal Evolution, Peking University. The laser beam was 36 µm and frequency was 10 Hz. Harvard zircon 91500 was used as an external standard for zircon U–Th–Pb analyses, and NIST610 as an external standard to calculate the contents of U, Th, Pb, and other trace elements. Zircon Lu–Hf isotopic analyses were conducted on the same or adjacent domains as the original pits used for LA–ICP–MS U–Pb isotopic analyses, using a Neptune Plus MC–ICP–MS attached to a Geolas 2005 excimer ArF laser ablation system at the state Key Laboratory of Geological Processes and Mineral Resources, China University of Geosciences in Wuhan. Beam diameter of 44 µm and repetition rate of 6 Hz were applied, and zircon 91500 and GJ-1 were used as the external standard and the unknown, respectively. The detailed analytical procedures for the above whole-rock chemical data, zircon U–Pb, and Lu–Hf isotopes are presented in Supplementary Information.

**Samples.** TTG gneisses are the dominant geological rock type in the Archean basement blocks of the North China Craton. Supplementary Figure 1 shows the general spatial and temporal distribution of Meso- to Neoarchean TTG gneisses, and detailed zircon age data and sampling locations are summarized in Supplementary Data 3. The magmatic crystallization ages of TTG gneisses are almost continuous from 3.0 to 2.5 Ga, with three peaks occurring at ~2.91, ~2.72, and ~2.53 Ga, with the youngest defining the maximum peak. The ~2.9 Ga TTG gneisses are sporadically distributed in Jiaodong, Anhui and Eastern Hebei terranes, whereas the ~2.7 Ga TTG gneisses concentrate more in the North Liaoning-South Jilin, Jiaodong, West Shandong, Dengfeng-Taihua, Zanhuang and Fuping terranes. The ~2.5 Ga data are distributed throughout the entire Eastern Block of

the North China Craton. All the TTG gneisses, irrespective of age exhibit similar petrographic characteristics, making the different age groups difficult to distinguish in the field. A detailed outline of the geological background of each terrane is given in the Supplementary Information. After carefully screening, only 155 samples of TTGs meet all the requirements, and we create further data subsets by grouping individual analyses with similar ages and localities (Supplementary Data 4–5). Our selected crustal-derived TTGs display low loss on ignition values of less than 2.0 wt %, and lack of significant Ce anomalies ($\delta Ce = Ce_N/\sqrt{(La_N \times Pr_N)}$ with values of 0.79–1.31, indicating that the rocks have not been subjected to any major alteration (Supplementary Data 4)[56]. Therefore, the chemical composition of the crustal-derived TTGs is considered to be close to that of the TTG magmas.

**Selection of mafic source rocks of the TTG magmatism.** Considering the isotopic and geochemical characteristics of crustal-derived TTGs, we collected Meso- to Neoarchean supracrustal rock data from the Eastern Block of the North China Craton and selected the low-K tholeiitic rock data among them (Fig. 2d, e). Calculations show that the average compositions of ~2.9, ~2.7, and ~2.5 Ga tholeiitic rocks are slightly different, especially the Mg# values, LOI and LILEs (such as Sr, Ba, and K) contents (Supplementary Data 6; Fig. 2f). Consequently, we take the average composition of each age low-K tholeiitic rock from the Eastern Block of the North China Craton as the potential source of contemporaneous TTGs, respectively.

**Thermodynamic modeling of primary magma.** Thermodynamic modeling was performed using the average composition of ~2.9, ~2.7, and ~2.5 Ga low-K tholeiitic rocks (Supplementary Data 6) from the Eastern Block of the North China Craton to better define the P–T conditions of the primary magma, following a Gibbs free energy minimization approach using the software Perple_X (version 6.9.0). Based on the mineral assemblages and bulk rock compositions, we choose a system $Na_2O–CaO–K_2O–FeO–MgO–Al_2O_3–SiO_2–H_2O–TiO_2–O_2$ (NCKFMASHTO) for the thermodynamic modeling, and use the hp633ver data set, assuming $Fe^{3+}/(Fe^{3+} + Fe^{2+}) = 0.1$[27,57,58]. Newly calibrated solution models are used for the melt, amphibole, and clinopyroxene[59]; garnet, orthopyroxene, ilmenite, and mica[60]; plagioclase[61] and biotite[62]. Quartz, rutile, titanate, and water were considered pure phases. The P–T phase diagrams and equilibrium mineral assemblages were calculated at discrete P–T points for every 10 °C and 0.1 GPa from 750 to 950 °C and 0.6 to 2.0 GPa, with water contents ranging from 1.2 to 2.0 wt% (Supplementary Data 7).

**Batch partial melting modeling.** Since the thermodynamic modeling yields the weight percentages of melts and equilibrium mineral assemblages, we can further calculate the trace element composition of the TTG melts through trace element modeling. Shaw[39] proposed the classical equation of batch partial melting

$$C_{melt}/C_{source} = 1/[D + F \times (1-D)] \tag{1}$$

Where $C_{source}$ and $C_{melt}$ represent the trace element concentration of the source rock and the resultant melt, respectively; $D$ is the bulk partition coefficient, and $F$ is the degree of partial melting (namely, the mass fraction of melt). The average composition of the ~2.9, ~2.7 and ~2.5 Ga low-K tholeiitic rocks from the Eastern Block of the North China Craton was used as the $C_{source}$, and the results ($C_{melt}$) were compared with the median composition of TTG melts with different ages of ~2.9 to ~2.5 Ga. The partition coefficient of minerals used in the batch partial melting modeling is from https://earthref.org/GERM/KDD/ (Supplementary Data 8).

**Geothermal models.** If the layer has thickness $\Delta z$, then the temperature at, and heat flow through, the bottom of the layer ($T_B$, $q_B$) can be expressed in terms of the temperature and heat flow at the top of the layer ($T_T$, $q_T$) and properties ($A$,$k$) of the layer[31].

$$T_B = T_T + \frac{q_T}{k}\Delta z - \frac{A\Delta z^2}{2k} \tag{2}$$

$$q_B = q_T - A\Delta z \tag{3}$$

Equations (2) and (3) are applied to successive layers, resetting $T_T$ and $q_T$ at the top of each new layer with the values $T_B$ and $q_B$ solved for the bottom of the previous layer. Where $A$ is volumetric heat production (W m$^{-3}$) and $k$ is thermal conductivity (W m$^{-1}$ K$^{-1}$). In practice, heat production and thermal conductivity are described by piecewise continuous functions and the computations are carried out with a 0.1 km depth increment. Following Tang et al.[63], the volumetric heat production in the crust is

$$A_j = \sum_j F_j \rho \sum_i H^i C_j^i e^{-\lambda^i t} \tag{4}$$

where $i = {}^{40}K$, ${}^{232}Th$, ${}^{235}U$ and ${}^{238}U$, $j$ indicates mafic and felsic crust, $F$ is volumetric proportion, $\rho$ is the density (2800 kg m$^{-3}$), $\lambda^i$ is the decay constant (yr$^{-1}$), $t$ is time before present (yr), $H^i$ is heat production (mW m$^{-3}$), $C^i$ is the present-day concentration (Supplementary Data 9). The average $K_2O$, Th and U concentrations in the Archean mafic and felsic crust are from Tang et al.[63], and the heat productivity of radioactive element is from Turcotte and Schubert[15]. The basic relation between heat

flow and the geothermal gradient is Fourier's law

$$q_B = -k \frac{\Delta T}{\Delta x} \qquad (5)$$

In this case, the $\frac{\Delta T}{\Delta x}$ represent the geothermal gradient slope (°C km$^{-1}$) of the Moho surface. Assuming that the lithospheric mantle does not produce heat ($A = 0$ W m$^{-3}$), we use a MATLAB solution to further constrain the variation trend of Archean continental geotherm. The MATLAB code was attached in Supplementary Code 1.

## Data availability

The authors declare that all data supporting the findings of this study are available online (https://doi.org/10.6084/m9.figshare.14644044.v1).

## Code availability

The MATLAB code used to reconstruct the geotherm model of the continental lithosphere is provided in a public repository 'figshare' with the identifiers (https://doi.org/10.6084/m9.figshare.14644044.v1).

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

## Acknowledgements

This research was supported by grants from the National Nature Science Foundation of China (41772188 and 41530207) and the Australian Research Council (FL160100168).

## Author contributions

S.L. designed the project and developed the initial idea. S.L., G.S., Y.H., and L.G. carried out field geological exploration and sampling. G.S. conducted the experimental analysis and simulations. F.H. compiled the data of Archean TTGs. M.T. and J.v.H. contributed to reconstructing the continental geotherm models. S.L., P.A.C., and G.S. wrote the paper. All authors participated in further discussions.

## Competing interests

The authors declare no competing interests.
