## [Peer Review File · Nature Communications]

Reviewers' Comments:

Reviewer #1:

Remarks to the Author:

I was very interested to read this paper, which presents data from low-Mg TTG that the authors claim constrains the thickness and geotherma gradient of Archean continental crust in the NCC, which they relate to changing geodynamic regimes driven by secular cooling of the mantle. There are some very interesting ideas in here, and the paper is both novel and provocative. It is based on assumptions about the composition of the metabasaltic crust from which the TTGs were derived, and on where within the crust they were derived, which are undoubtedly simplifications.

In essence, I am not convinced that the authors are correct, but they might be. Although I don't agree with everything in it, I can find nothing fundamentally wrong with it. In any case, the paper will certainly generate discussion and debate on a topic that remains of broad interest to many, which is very welcome.

I've noted a few points below that the authors might like to think about.

25 – It's not entirely clear why 'tonalitic-trondhjemitic-granodioritic gneisses are ideal lithologies for reconstructing the thermal state of early continental crust'. From a phase equilibrium perspective, they generally contain high-variance assemblage stable over large P-T fields.

54 – 'partial melting of fluid fluxed mantle' does not have to invoke slab melting. See the paper by Smithies et al. (No evidence for high-pressure melting of Earth's crust in the Archean, Nature communications 10 (1), 1-12), on the origin of high-P TTGs by shallow melting of enriched lithosphere.

57 – 'with Mg# values ($Mg\# = Mg^{2+}/(Mg^{2+} + Total\ Fe^{2+}) \times 100$) lower than 45 based on the results of experimental petrology²⁷.' The modelling of Johnson et al. ('Earth's first stable continents did not form by subduction, Nature 543, 239-242) suggest a protolith with Mg# as low as 35. This might make a useful complementary piece of evidence, unless you are talking about the composition of the TTGs (melts), which perhaps you are, in which case ignore me. In addition, there is no need to add the charge of magnesian as it is implied.

61 – yes, but there have been several papers of late questioning the robustness of the pressure estimates that probably deserve to be discussed briefly.

94 – A picky point, but I'm not sure the ref. 22 is 'the classical definition of Archean TTGs'. Many 'definitions' exist.

97 – Importantly, there is a big overlap in compositions between the high- and low-Mg TTGs. On the basis of which composition criteria were they separated (into blue squares and red triangles)? This needs to be clearly stated up front.

122 – Purely as a comment, the term 'Archean 'arc-like' basalts' really should be expunged. It is loaded and very unhelpful.

Fig. 2d – The estimates of proportion of garnet in the source need to be better explained here. While extremely useful, the estimates of Moyen (2011) are rather simplistic. For example, the amount of garnet is strongly dependent on the Mg# of the metabasaltic source (see Johnson et al. 2017), not only depth. I guess if you assume that, on average, the source composition was the same in the different areas (any evidence for this?), then this might be OK, but this assumption needs to be more clearly stated.

Fig. 3. To labour the point, the estimates of crustal thickness rely (I think) on: (i) a single (or same on average) source composition, and; (ii) that the TTGs were derived from the very base of the crust. Are these assumptions justified and are they consistent with secular cooling over 0.4 Ga?

195 – As a mini rant, I also dislike the term 'the mantle plume regime', or at least its implications to many. I am not at all convinced that anomalously high mantle T in the Archaean is evidence for plumes as most people think about them (i.e. originating at the core–mantle boundary). I think they may have been generated within the mantle transition zone or reflect thermal anomalies at even higher levels, as many others have proposed (I think). I'm also not that keen on 'hot subduction' but I may be in a minority.

How do your findings based on samples from the Dengfeng complex fit with the recent paper by Huang et al. (Paired metamorphism in the Neoproterozoic: A record of accretionary-to-collisional orogenesis in the North China Craton, EPSL 543, 116355)?

Finally, spending a few hours perfecting the English (which is fine) would make this even more readable and is probably worth the effort.

Tim Johnson,

Perth, Oct. 2020.

Additional references to consider:

Brown et al. Plate tectonics and the Archean Earth, Annual Review of Earth and Planetary Sciences 48, 291–320.

Reviewer #2:

Remarks to the Author:

When the plate tectonics launched on Earth is hereto hotly debated. This paper reports the Meso-Archean (ca. 3.0 Ga) switch of thermal state in the lower crust or Moho surface archived by the low Mg# TTGs in the Eastern block of North China Craton (NCC). The derivation results suggest the colling of thermal gradients of Moho and thickening of crust resulted from hot subduction during the Meso- to Neo-Archean time. The authors state that this phenomenon probably typifies the transition of geodynamic regimes from divergent to convergent, as a function of modern-like plate tectonics (hot subduction). This contribution gives a very valuable insight into the evolving plate tectonics from the perspective of switch of the lithospheric thermal structure.

However, I still have concerns about the derivation procedure from the basic data to the findings and therefore recommend a substantial major revision of this manuscript, after considering the reservations below, before publication.

(1) Generally, the value of Mg# typifies the degree to which the mantle proportion interacts and indicates the source of magma. It has been estimated that the bulk composition of the continental crust has Mg # of 34 to 42 (%). Here why did the authors select the value '45' as the dividing line between the magmas originating from the crust and the mantle (in Lines 54 to 60). In fact, this value was not, in particular, expressly stated in the cited papers as a magmatic source boundary (Rapp et al., 1999; Rapp and Watson, 1995; Stern and Kilian, 1996). On the contrary, Stern and Kilian (1996)'s work suggested when subducted oceanic basalt served as the source instead of the lower crust, the Mg# of the Austral Volcanic rocks (adakites) could be low to 40. Could the authors further explain the

Mg# '45' option? If necessary, a plot or diagram illustrating the distribution of Mg# of all collected samples is better (more legible for readers) to set forth.

(2) What is the application condition on the use of Sr/Y and La/Y indicators for reconstructing crustal thickness? Based on the recent work (Figure 1a in Hu et al., 2020), the regression estimation of crustal thickness commonly produced larger errors due to data discretization in the thin crustal areas (~10-40 km), much less reliable than the thicker ones. How to judge or avoid the error when estimating thin crustal thickness by using the geochemical indicators, for instance when estimating the ~30-40 km Meso-Archean crustal thickness in this contribution.

(3) In addition, the geochemical modeling only took the batch partial melting into account. Is the only paragenesis of the TTGs in NCC? How to exclude magma mixing or assimilation and fractional crystallization (AFC) processes? Actually, some research progresses giving birth to new interpretations (models) that challenged the conventional consideration for the Mg# and generations of alleged 'HP' to 'LP' TTGs, which stressed melt segregation and differentiation of a single magma source in the continental crust (Getsinger et al., 2009; Laurent et al., 2020). How do the authors consider the paradox for the genesis of TTGs?

(4) A cooling trend at the lower continental crust from the Meso- to the Neo-Archean was unraveled in this contribution through numerations of basal heat flow and metamorphic phase pseudosection modeling. From a practical perspective, was the temperature fall recorded by the rock or minerals in TTGs? Evaluating the crystallization temperatures of TTGs in the NCC via geo-thermometry, such as Ti-in-zircon and zircon saturation geothermometers, may provide straightforward mineralogical proof for the temperature drop or cooling trend.

(5) The proportion of the residual phase basing the geochemical modeling were given to be some certain values (in Lines 127-133). However, the plots in Figures 2d-2f show ranges (or errors) of the residual proportion of garnet, instead of certain values, at each sampling locality. Do the ranges (or errors) affect much the pseudosection modeling result or not? Perhaps, this is the reason for the inconsistency between the thermodynamic modeling and the basal heat flux calculation. For instance, the thermal condition generating the ~2.7 Ga Zhongtiao TTG (in Figures 5a & 5b) is shown as ~16 °C/km and the temperature can be deduced to be ~720 °C (depth*thermal gradient). While this sample is plotted at the conditions of ~900 °C and ~20 °C/km in the pseudosection by the contouring Sr/Y and garnet proportion (Figure 3). Furthermore, why were only 3 to 6 samples selected for plotting in pseudosection (Figure 3) and estimating thickness and thermal gradient (Figure 5)?

Specific comments:

(1) Line 49, 'constituting' should be corrected as 'constitutes'.

(2) Line 65, 'quantitative' should be corrected as 'quantitatively'.

(3) Line 227, necessary and representative citation on the orogenic cycle of subduction in the Eastern Block during the early to late Neoproterozoic should be inserted behind.

(4) Line 461 in the bibliography, Chapman (1986)'s paper was published in the <Geological Society, London, Special Publications>.

References:

Getsinger, A., Rushmer, T., Jackson, M.D., Baker, D., 2009. Generating high Mg-numbers and chemical diversity in tonalite-trondhjemite-granodiorite (TTG) magmas during melting and melt segregation in the continental crust. *J. Petrol.* 50, 1935–1954.

<https://doi.org/10.1093/petrology/egp060>

Hu, F., Wu, F., Chapman, J.B., Ducea, M.N., Ji, W., Liu, S., 2020. Quantitatively Tracking the Elevation of the Tibetan Plateau Since the Cretaceous: Insights From Whole-Rock Sr/Y and La/Yb Ratios. *Geophys. Res. Lett.* 47, 1–10. <https://doi.org/10.1029/2020GL089202>

Laurent, O., Björnson, J., Wotzlaw, J.F., Bretscher, S., Pimenta Silva, M., Moyen, J.F., Ulmer, P., Bachmann, O., 2020. Earth's earliest granitoids are crystal-rich magma reservoirs tapped by silicic eruptions. *Nat. Geosci.* 13, 163–169. <https://doi.org/10.1038/s41561-019-0520-6>

Rapp, R.P., Shimizu, N., Norman, M.D., Applegate, G.S., 1999. Reaction between slab-derived melts and peridotite in the mantle wedge: Experimental constraints at 3.8 GPa. *Chem. Geol.* 160, 335–356. [https://doi.org/10.1016/S0009-2541\(99\)00106-0](https://doi.org/10.1016/S0009-2541(99)00106-0)

Rapp, R.P., Watson, E.B., 1995. Dehydration melting of metabasalt at 8-32 kbar: Implications for

continental growth and crust-mantle recycling. *J. Petrol.* 36, 891–931.

<https://doi.org/10.1093/petrology/36.4.891>

Stern, C.R., Kilian, R., 1996. Role of the subducted slab, mantle wedge and continental crust in the generation of adakites from the Andean Austral Volcanic Zone. *Contrib. to Mineral. Petrol.* 123, 263–281. <https://doi.org/10.1007/s004100050155>

Reviewer #3:

Remarks to the Author:

Please see the attached file. I am happy to waive anonymity here and am open to the authors contacting me to clarify any points, if needed.

Review of: “Thermal state and evolving geodynamic regimes of the Meso- to Neoproterozoic North China Craton”

This manuscript details changing crustal (lithospheric) thickness across the North China Craton over a period of c. 2.9 Ga to c. 2.5 Ga, and the authors then examine whether these reported changes could have been driven by a transition from stagnant lid to mobile lid tectonics, as this time period is noted by many people as being a strong candidate for the timing of onset of global subduction. It does so by using a two-pronged approach. First, by analyzing the geochemistry and age of crustal-derived TTG gneiss from across the craton, with the idea that the maximum pressure of magma generation can constrain the maximum thickness of the continental crust at this time. Then, these pressures are used to build a geothermal model that varies in space and time. I have no issue with the methods and results of this second stage, but I have moderate to major concerns about the validity of assumptions and interpretations made as they relate to this first stage (i.e. working out “pressure” of formation of the TTG parent magma). One critically flawed assumption is that the same generic protolith can be used to calculate the garnet content of the residue from which these TTG magmas were derived. Many studies have shown that the metamorphic parageneses in the residue (and so the volume of melt and its composition generated) are highly sensitive to protolith composition. Further, I see no convincing evidence that these TTG have to have been derived at the base of over-thickened crust. The geochemical arguments for this are weak (and non-specific), and some of the sample set could equally have been derived via shallow subduction, which would give the impression of a thick overlying crust, where one did not exist. Finally, the “matching” of trace element ratios against interpreted garnet vol. % in the residue is shaky at best, and unfortunately getting robust values for pressure of magma generation is a cornerstone of the subsequent thermal modelling. In principle, this approach is innovative and should lead to a high-impact study that would be of much interest to many workers worldwide studying Archean geodynamics. However, there are too many points of contention for me to recommend publication in its current form. I hope the authors find these comments useful.

Line 37: Continental or oceanic lithosphere, or both? I’d like to see the writing tightened up considerably throughout to remove ambiguity such as this.

Lines 40–42: Please clarify (and/or simplify) this for the general reader. What exactly do you mean by “transition between different thermal states”? A decrease in mantle T_p ? Conduction vs. convection as the dominant mode of heat transport?

Line 46: Missing word (or letter?) at the end of this sentence.

Line 47: Is ‘basement’ the right word here? Greenstones are not ‘basement’ lithologies. If you go to northwest Scotland, for example, you find high-grade amphibolites and mafic granulites, which form the true basement of Archean terranes.

Lines 47–49: Where does this value of 70 vol. % come from? The cited Martin et al. (2005) paper? More recent studies suggest that the Archean crust was more mafic than previously thought (e.g. Tang et al., 2016), with TTG gneiss making up subsidiary amounts, although that study does explicitly focus on the uppermost crust.

- Tang, M., Chen, K. and Rudnick, R.L., 2016. Archean upper crust transition from mafic to felsic marks the onset of plate tectonics. *Science*, 351(6271), pp.372-375.

Lines 54–55: This is a mistake, I assume? A slab melting model involves melting of a subducted slab (i.e. the basaltic portion of subducted oceanic lithosphere) – not devolatilization of subducted crust, infiltration of this fluid into the mantle wedge, and melting thereof. None of the papers cited here (refs. 24 to 26) suggest that fluid fluxing into the mantle creates TTG magmas. In any case, there are two distinct end-members of the slab melting hypothesis: shallow and steep (figure below from Palin et al., 2016):

Shallow slab melting becomes very important for your discussion/sample selection later, because the lack of an intermediate mantle wedge makes TTG melts derived in those scenarios resemble melts formed at the base of a crustal column (i.e. they are not affected as much by interaction with the ultramafic mantle during ascent). I'd advise rewording this discussion of "end members" appropriately.

- Palin, R.M., White, R.W. and Green, E.C.R., 2016a. Partial melting of metabasic rocks and the generation of tonalitic–trondhjemitic–granodioritic (TTG) crust in the Archaean: Constraints from phase equilibrium modelling. *Precambrian Research*, 287, pp.73-90.

Line 62, 66, etc: It's absolutely critical to tighten up the nomenclature here. The TTG gneisses are not the same as the TTG magmas generated from melting of a meta-basalt. The latter comes first and is later transformed into the former. Everywhere that you refer to TTG gneiss (incorrectly) you need to refer to the melt instead.

Line 90–91: I am highly dubious about this statement and division. Given that only the low-MgO data are presented in supplementary information, it's not easy for me to check the rigor of this sample selection. Further, it seems from Fig. 1 that there is **considerable** overlap in geochemistry and mineralogy between both groups. It's evident that symbols for each overlap in broad regions. A heat map or adding some kind of transparency would be much more helpful to allow the reader to see how distinct they are in the region of

commonality. The absolute values of MgO, for example (lines 98 and 102) do not seem sufficient to distinguish the two.

I understand that there is a desire to filter the dataset to only include magmas generated via vertical burial and crustal thickening, but I think the authors need to do a much better job of eliminating the possibility that these melts could have formed in a shallow subduction zone (see figure above), which phase equilibrium modelling has shown is a viable geodynamic environment to form TTGs of appropriate mineralogy and chemistry matching natural Archean terranes (i.e. Palin et al., 2016a, and many others).

Line 118: Some data fall in the eclogite field of these discrimination diagrams? This gives more support for a shallow subduction environment, at least for some of your data. You would need exceptionally thick mafic crust to generate an eclogite base (instead it's more likely to be mafic granulite).

Lines 121–123: Arc basalts/tholeiites are extrusive rocks – is there a thought as to how they can end up as the juvenile source rocks that melt at lower-crustal pressures (depths) to form these voluminous TTGs?

Lines 126–133: The line-fitting precision for these supposed volumes of garnet in the residue is extremely poor. The examples given for East Hebei and Anhui are both based on two points, none of which lie on the 10% garnet curve. The example for Jiaodong seems like it would fit considerably better with 25% or 30% garnet, not 20%. It would be easier to see these data if all lines with 5% increments were shown on each diagram. Was there any analytical calculations performed here to determine garnet volume, or was it simply curve fitting?

Aside from this poor fit being a source of uncertainty, this analysis relies entirely on the assumption that the generic arc basalt used as the protolith is (a) correct and (b) chemically homogenous for all examples. Is it believable that the entire base of the NCC can be described by a single, chemically homogenous source rock?

Lines 133–136: This is just too broad of a statement/assumption. There are several variables that could account for this – not solely pressure – that are not justified. Many studies using experimental petrology and/or phase diagram analysis show that garnet stability is also temperature and bulk composition dependent. Presumably you can see this for yourself in the diagram calculated for Figure 3?

Line 139: I see in the Methods that you used a CIPW norm assessment to determine this water content. If you're using the bulk composition of Archean arc-like basalt from Table S5 for the rest of the modelling, why not just use the (measured) average LOI? Isn't it more reliable than introducing unnecessary assumptions when dealing with the normative anhydrous results?

Lines 140–142: This short sentence touched on a conceptual issue that I raised before. The TTG magmas are not the same as the TTG gneisses. How do you take into account the magmas separating from their source, ascending through the crust, stalling, and crystallizing? Is this change in pressure significant? Could trace element ratios have been changed during magma ascent, thus calling into question your prior assumptions of garnet proportion in the residue?

Lines 143–145: Given the many intrinsic sources of uncertainty in phase diagram modelling (e.g. Powell and Holland, 2008), it's really not appropriate to report calculated linear geotherms with a precision of one decimal place. Just simplify them (e.g. 18.2–22.8 °C/km -> 18-23 °C/km). To be completely rigorous, you probably should take into account the standard uncertainty on P-T estimation using phase equilibrium modelling (± 50 °C and ± 1 kbar at 2 sigma: Palin et al., 2016b).

- Powell, R. and Holland, T.J.B., 2008. On thermobarometry. *Journal of Metamorphic Geology*, 26(2), pp.155-179.
- Palin, R.M., Weller, O.M., Waters, D.J. and Dyck, B., 2016. Quantifying geological uncertainty in metamorphic phase equilibria modelling; a Monte Carlo assessment and implications for tectonic interpretations. *Geoscience Frontiers*, 7(4), pp.591-607.

Lines 158–159: What are these thicknesses when you take into account uncertainty in pressure estimation (cf above)? What was your pressure-to-depth conversion ratio? Was it constant throughout the crust? If so, why, as the overburden likely changes with absolute depth (i.e. upper crust is more volcanic and potentially mafic greenstone, so has a different density to the felsic pluton-rich middle crust)? You are hanging your hats on these changes in depth, so the reader should be fully briefed on the uncertainties on the numbers and how they were derived.

Line 189: This is not “relatively thin” compared to modern-day average continental crust and in my mind further supports the possibility that some TTG magmas were generated at depths exceeding that of the crust (i.e. in a shallowly subducted slab). Have you considered how these numbers weigh up against independent estimates of Archean continental crust thicknesses (e.g. Dhuime et al., 2015)?

- Dhuime, B., Wuestefeld, A. and Hawkesworth, C.J., 2015. Emergence of modern continental crust about 3 billion years ago. *Nature Geoscience*, 8(7), pp.552-555.

Lines 209–217: This is the first mention I can see of possible shallow subduction. This must be discussed much earlier in the manuscript.

Lines 255–256: I’m sorry to keep bringing this up, but subsequent studies have shown that this tripartite classification simply leads to confusion. The figure on the next page (again from Palin et al., 2016a) shows mineral proportions in the residue (specifically garnet and plagioclase) and melt fraction generated from an enriched Archean tholeiite – like your example – at a range of P-T conditions appropriate for TTG generation. Garnet proportions of >30 vol. % can be generated at only moderate pressures (~12-14 kbar = 36-42 km depth) at only moderate temperatures (~850-900 °C). All of Moyen’s work is of exceptional quality, but the advent of petrological modelling and forward calculation of trace element ratios and residual mineral assemblages can demonstrate easily that high-pressures (>20 kbar, according to the early studies, e.g. Rapp et al., 1990) are not required to generate a TTG with a “high-pressure” signature. This is just a fact, and I assume that you accept the value and reliability of this kind of thermodynamic modelling, as you employ it in your own study here.

Methods

Line 319: What rock? Your bulk composition is a statistical average, isn’t it? How do you know the source rocks don’t contain apatite? If they don’t, where does the P₂O₅ reside?

Lines 323–327: This must be an error, because the Green et al. (2016) a-x relations are not compatible with the pre-2011 Holland and Powell dataset. Can you confirm this?

Lines 560–563/Fig. 3: The contours (dashed lines) for garnet % and Sr/Y in the melt cannot be correct, and I am always hesitant to make such statements for phase diagrams that I have not produced, but there are clear problems with them. I have enough experience producing these pseudosections to know that the slopes of isopleths that are controlled by intensive variables (i.e. mass of a phase) change across field boundaries. See

the garnet volume contours on the figure from Palin et al. (2016a), next page, as an example. For instance, one of the garnet contours marked here for 15 vol. % intersects the garnet-in field boundary. This cannot be true – there cannot be simultaneously 10 vol. % and 0 vol. % of garnet at the same P-T condition (~0.9 GPa and ~910 °C). Secondly, how were the Sr/Y ratios calculated? Why don't they show similar trends to the garnet proportions as you are in a closed (bulk composition constrained) system? On that note, why is melt loss not considered here? It is thought that melt escapes in pulses once a critical threshold is reached. There are too many small issues with this phase diagram for me to be confident that the calculated data (i.e. pressures) are reliable.

Supplementary Information

Fig S11 – What are the grey ellipses on these plots? Please expand the caption.

Selection of mafic source rock – this result (“Geochemical modeling results clearly show that most MORBs and OPBs are too depleted in LILEs to generate Archean TTGs, whereas the IABs are slightly too enrich. The

most likely mafic source rock for Archean TTGs is Archean 'arc-like' basalt) has been rigorously demonstrated previously by Martin et al. (2014). Why was this treatment repeated here?

- Martin, H., Moyen, J.F., Guitreau, M., Blichert-Toft, J. and Le Pennec, J.L., 2014. Why Archaean TTG cannot be generated by MORB melting in subduction zones. *Lithos*, 198, pp.1-13.

Richard Palin

Associate Professor of Petrology, University of Oxford

Response to referees letters

Response to comments of Reviewer #1 (Prof. Tim Johnson)

I was very interested to read this paper, which presents data from low-Mg TTG that the authors claim constrains the thickness and geothermal gradient of Archean continental crust in the NCC, which they relate to changing geodynamic regimes driven by secular cooling of the mantle. There are some very interesting ideas in here, and the paper is both novel and provocative. It is based on assumptions about the composition of the metabasaltic crust from which the TTGs were derived, and on where within the crust they were derived, which are undoubtedly simplifications.

In essence, I am not convinced that the authors are correct, but they might be. Although I don't agree with everything in it, I can find nothing fundamentally wrong with it. In any case, the paper will certainly generate discussion and debate on a topic that remains of broad interest to many, which is very welcome.

I've noted a few points below that the authors might like to think about.

25 – It's not entirely clear why 'tonalitic–trondhjemitic–granodioritic gneisses are ideal lithologies for reconstructing the thermal state of early continental crust'. From a phase equilibrium perspective, they generally contain high-variance assemblage stable over large P – T fields.

Thank you for your comment. It is generally agreed that tonalitic–trondhjemitic–granodioritic (TTG) gneisses were mostly produced by partial melting of hydrated mafic rocks at depth corresponding to at least 0.8 to 1.0 GPa. However, in what geodynamic environment this deep melting occurred is highly controversial, resulting in the TTG melts contain high-variance assemblage stable over large P – T fields in the P – T phase diagram. Among them, a part of the TTGs formed near the base of thickened crust, their melt composition is dominantly controlled by mafic source compositions and melting conditions (i.e., pressure and temperature). Therefore, the Archean crust-

sourced TTG gneisses are ideal lithologies for reconstructing the thermal state of early continental crust. To clarify this, we added the current genesis research status of Archean TTGs and provided a systematic petrogenetic classification of TTGs in the new version of the manuscript (lines 50–56 and 101–122).

54 – ‘partial melting of fluid fluxed mantle’ does not have to invoke slab melting. See the paper by Smithies et al. (No evidence for high-pressure melting of Earth’s crust in the Archean, *Nature communications* 10 (1), 1-12), on the origin of high-P TTGs by shallow melting of enriched lithosphere.

This is now revised. The ‘melting of fluid fluxed mantle’ is indeed different from ‘slab melting’. According to your suggestion, we have complemented the petrogenetic classification of TTGs as following: (1) fractional crystallization of mafic melts from an enriched lithospheric mantle (Bai et al., 2014; Smithies et al., 2019; Laurent et al., 2020); (2) partial melting of hydrated mafic rocks at the base of continental crust (Moyen and Martin, 2012; Johnson et al., 2017; Fig. 1a); and, (3) melting of subducted oceanic slabs (including steep and shallow subduction) (Palin et al., 2016a; Fig. 1b, c). See lines 50–56, please.

57 – ‘with Mg# values ($Mg\# = Mg^{2+}/(Mg^{2+} + Total\ Fe^{2+}) \times 100$) lower than 45 based on the results of experimental petrology²⁷.’ The modelling of Johnson et al. (‘Earth’s first stable continents did not form by subduction, *Nature* 543, 239–242) suggest a protolith with Mg# as low as 35. This might make a useful complementary piece of evidence, unless you are talking about the composition of the TTGs (melts), which perhaps you are, in which case ignore me. In addition, there is no need to add the charge of magnesian as it is implied.

Revised. Thank you for sharing the Nature’s paper about the formation of Earth’s first stable continental, which took the Coucal basalt with an Mg# of 35 as the suitable source for the East Pilbara TTGs. However, we do not mean that the basaltic protolith of the TTG melt has Mg# values below 45, but that the TTG melt itself has Mg# value

below 45 in our original text. The division of both high-Mg and low-Mg TTGs was not based on a specific Mg# value, but on the distribution characteristics of Mg# in experimental petrology studies (Fig. 2c; Rapp et al., 1999; Rapp and Watson, 1995; Stern and Kilian, 1996). Therefore, we deleted the original confusing expression and revised the classification scheme for TTGs. Furthermore, we also removed the charge of magnesium from the formula according to your advice. See lines 64–65 and 112–120, please.

61 – yes, but there have been several papers of late questioning the robustness of the pressure estimates that probably deserve to be discussed briefly.

Revised. Thanks for your suggestion here. We have updated the text related to ‘pressure estimation of TTG melts’. Moyen (2011) classified the Archean (~3.5–2.5 Ga) TTGs into three different categories (high-pressure, medium-pressure and low-pressure subgroups) based on their geochemical indicators (i.e., Sr contents, Sr/Y and La/Yb ratios). Most TTGs (~80 %) belong to the low-pressure (LP) and medium-pressure (MP) subgroups that originated from thickened lower crust (around 30–45 km depth), whereas the remaining 20 % are high-pressure (HP) TTGs that likely derived from subduction zones (> 60 km). However, this pressure estimation scheme ignores the complexity of petrogenetic process and source rock composition, which has been questioned by many scholars. Johnson et al. (2017) demonstrated that the amount of garnet in residue is strongly dependent on the Mg# values of the metabasaltic source rocks, and propose that MP TTGs may be stable under the pressure of 0.7 GPa (~25 km). Furthermore, Smithies et al. (2019) suggested that HP TTGs were not derived from partial melting of crustal materials, but from fractional crystallization of mafic melts that originated from metasomatically enriched lithospheric mantle. Above all, the TTGs’ geochemical features (i.e., Sr/Y and La/Yb ratios) do not only depend on the pressure (P) –temperature (T) conditions generating the TTG melts, but also on the source compositions and petrogenetic processes. See lines 56–70, please.

94 – A picky point, but I’m not sure the ref. 22 is ‘the classical definition of Archean TTGs’. Many ‘definitions’ exist.

Revised. Thank you for your suggestion, we have rewritten the original expression into “We define TTGs as silica-rich granitoids ($\text{SiO}_2 > 64$ wt.%) with high Na_2O ($3.0 \text{ wt.}\% \leq \text{Na}_2\text{O} \leq 7.0 \text{ wt.}\%$) and Al_2O_3 ($\text{Al}_2\text{O}_3 > 13$ wt.%) contents (Moyen and Martin, 2012)”. See lines 97–98, please.

97 – Importantly, there is a big overlap in compositions between the high- and low-Mg TTGs. On the basis of which composition criteria were they separated (into blue squares and red triangles)? This needs to be clearly stated up front.

Revised. Thank you for your suggestion, we summarize a large amount of experimental petrology data (Supplementary Table S1) and modify the classification method for TTGs. In the SiO_2 versus Mg# diagram, the TTGs can be clearly divided into two groups based on the distribution characteristics of the Mg# values (Fig. 2c). The low-Mg TTGs exhibit relatively low MgO contents and Mg# values, and plot in the compositional range of experimentally obtained partial melts of amphibolite and eclogite. Whereas the high-Mg TTGs display higher MgO contents and Mg# values than crustal partial melts obtained from experiment. See lines 112–120, please.

122 – Purely as a comment, the term ‘Archean ‘arc-like’ basalts’ really should be expunged. It is loaded and very unhelpful.

Revised. Extant experimental and petrological studies show that the Archean moderately enriched tholeiitic basalts are ideal protolith for TTGs (Martin et al., 2014). Therefore, we have changed “Archean ‘arc-like’ basalts” to “enriched Archean tholeiites” in the updated text. See lines 123–124, please.

Fig. 2d – The estimates of proportion of garnet in the source need to be better explained here. While extremely useful, the estimates of Moyen (2011) are rather simplistic. For example, the amount of garnet is strongly dependent on the Mg# of the metabasaltic

source (see Johnson et al. 2017), not only depth. I guess if you assume that, on average, the source composition was the same in the different areas (any evidence for this?), then this might be OK, but this assumption needs to be more clearly stated.

Revised. As the reviewer states, the garnet proportion in the residue is strongly dependent on the Mg# values of metabasaltic rocks, not only the melting depth. Importantly, the composition of the protolith rocks (enriched Archean tholeiites) may be different at different times. Thus, we collected Meso- to Neoproterozoic supracrustal rock data from the Eastern Block of the NCC and selected the low-K tholeiitic rock data. Calculations show that the average compositions of ~2.9 Ga, ~2.7 Ga and ~2.5 Ga tholeiitic rocks are slightly different, especially the Mg# values, LOI and LILEs (such as Sr, Ba and K) contents (Supplementary Table 6; Fig. 2f). Based on the above analysis, our original estimation of garnet proportion in the residue was too simple to be consistent with the geological evidence, and therefore, we have deleted the original Fig. 2d–f and relevant descriptions. In the updated text, we carried out thermodynamic modelling using the latest determined mafic source rocks (for the time intervals ~2.9 Ga, ~2.7 Ga and ~2.5 Ga low-K tholeiitic rocks in the North China Craton). The specific weight percentages of melts and equilibrium mineral assemblages were calculated at discrete P – T points for every 10 °C and 0.1 GPa from 750° to 950 °C and 0.6 to 2.0 GPa, with a water contents ranging from 1.2 to 2.0 wt.% (Supplementary Table 7). The black and blue dashed contours in the P – T phase diagrams represent the calculated degree of melting (wt.% of melt) and garnet proportion (%) in residue, respectively (Fig. 3a, c and e).

Fig. 3. To labour the point, the estimates of crustal thickness rely (I think) on: (i) a single (or same on average) source composition, and; (ii) that the TTGs were derived from the very base of the crust. Are these assumptions justified and are they consistent with secular cooling over 0.4 Ga?

Revised. We very much appreciate the reviewer's point about estimations of crustal thickness. In the original manuscript, we took the median composition of enriched

Archean tholeiites as a single source for the TTGs with different periods, and considered all of the low-Mg TTGs ($Mg\# < 45$) were derived from partial melting of metamorphic mafic rocks in the lower crustal environment. These assumptions played a considerable part in estimating the ancient crustal thickness. Based on the reviewer's advice, we rechecked the chemical data of mafic source rocks and TTGs that were used for determining crustal thickness. Firstly, we collected Meso- to Neoproterozoic supracrustal rock data from the Eastern Block of the NCC and selected the low-K tholeiitic rock data among them. Calculations show that the average compositions of ~2.9 Ga, ~2.7 Ga and ~2.5 Ga tholeiitic rocks are slightly different, especially the $Mg\#$ values, LOI and LILEs (such as Sr, Ba and K) contents (Supplementary Table 6; Fig. 2f). Therefore, we took the average composition of each age group of low-K tholeiitic rocks as the potential source of contemporaneous TTGs. Subsequently, we undertook a systematic classification of the Meso- to Neoproterozoic TTGs from the Eastern Block of the NCC before discussing the crustal thickness, and the screening procedure consists of four steps: (1) We filtered data by SiO_2 (>64 wt.%), Al_2O_3 (>13 wt.%) and Na_2O (3.0–7.0 wt.%) according to the TTG's definition (Moyen and Martin, 2012), excluding non-typical TTG samples; (2) We removed TTG samples with heterogeneous whole-rock Nd and zircon Hf isotopic compositions, which may be derived from magma mixing or from contaminations of felsic crust as well as mantle peridotite during TTG magmatism; (3) We determined which TTG samples that were derived from partial melting process through the petrogenetic discrimination; (4) Based on the experimental results of basalts, we removed the high-Mg TTG samples that may be affected by mantle materials. After carefully screening, only 155 samples of TTGs (now including additional data published last year) meet the above criteria, and we create data subsets by grouping individual analyses with similar ages and geographic locations. In the updated text, we carry out thermodynamic and trace element modelling using the latest ~2.9 Ga, ~2.7 Ga and ~2.5 Ga mafic source rocks in the Eastern Block of the NCC and crustal-derived TTGs. The mineral proportion of residual phases and melting degree were obtained under specific $P-T$ conditions by thermodynamic calculations, then, we

simulated multiple trace element compositions of the partial melts based on the mineral proportion in the residual phases, melting degree and partition coefficient of trace elements using a simple batch partial melting model (Shaw, 1970), up to the chemical composition of modeled melts close to those of actual TTG samples, thus we could quantitatively calculate the pressure range of TTG melts, namely, the ancient crustal thickness. See the updated Fig. 3, please.

195 – As a mini rant, I also dislike the term ‘the mantle plume regime’, or at least its implications to many. I am not at all convinced that anomalously high mantle T in the Archaean is evidence for plumes as most people think about them (i.e. originating at the core–mantle boundary). I think they may have been generated within the mantle transition zone or reflect thermal anomalies at even higher levels, as many others have proposed (I think). I’m also not that keen on ‘hot subduction’ but I may be in a minority. Revised. I agree with the reviewer. People have suggested a range of regimes that are different from the modern plate tectonic regime: stagnant lid, dripping, sagduction, etc. I am quite convinced that the rocks refer to just two regimes: subduction-related melts, and plume-related melts. With the hotter mantle temperatures, melts can form in a much wider range of ways. In general, hot spots are the direct reflections of the mantle plume at Earth’s surface, marked by ultramafic volcanism, high heat flow and uplift (Wilson, 1973). Three lines of evidence strongly support that a plume-related vertical tectonic regime played a dominant role in the late Mesoarchean: (1) the late Mesoarchean komatiites in west Shandong exhibit Al-depleted geochemical compositions, which could form by melting in the high-temperature axis of the plume (Polat et al., 2006); (2) the granite-greenstone belt in the Eastern Block experienced ~2.85–2.80 Ga extensional deformation, which may be caused by a mantle upwelling (Gao et al., 2019); (3) high-mantle potential temperatures (T_p) in the late Mesoarchean has been suggested to favor cratonic mantle formation by plumes (Lee et al., 2011; Herzberg, 2016). Furthermore, thermomechanical numerical modellings show that the subduction velocity increase, the subducted angle steepen, and the subducted oceanic slabs were subjected to frequent

slab roll-back and drip-off when the mantle potential temperature is $\sim 100\text{--}200$ °C higher than modern (i.e., van Hunen and van den Berg, 2008). Therefore, we call it ‘hot subduction’ to distinguish from present-day cold subduction. See lines 201–208, please.

How do your findings based on samples from the Dengfeng complex fit with the recent paper by Huang et al. (Paired metamorphism in the Neoproterozoic: A record of accretionary-to-collisional orogenesis in the North China Craton, *EPSL* 543, 116355)? Revised. Huang et al. (2020) documented a Neoproterozoic (ca. 2.52–2.50 Ga) paired metamorphic belt in the Dengfeng complex, and estimated the peak or retrograde P – T conditions of Archean samples using mineral compositions in conjunction with THERMOCALC and conventional geothermobarometers, which is an excellent research result. The P – T conditions recorded by the metabasalt ($P = 6.3\text{--}10$ kbar, $T = 675\text{--}750$ °C) and tonalitic gneiss ($P = 5.6\text{--}8.8$ kbar, $T = 750\text{--}810$ °C) from the western part of the Dengfeng Complex were converted to a high apparent geothermal gradient (> 23 °C/km, defined as the direct temperature (T)/pressure (P) at the metamorphic peak). We understand that the P – T conditions and converted geothermal gradients should be varied at the evolutive stages of an orogenic cycle, thus the converted geothermal gradient is essentially an apparent geothermal gradient, which cannot represent the P – T conditions and geothermal gradients at the crust-mantle boundary. Furthermore, the peak P – T conditions and apparent geothermal gradients can be affected by multiple factors, for example, the local dynamic process of magmatic emplacement, fluid actions and structural/tectonic extension, breaking off, stacking or others. In this study, we didn’t calculate the apparent geothermal gradient at each stage of the orogenic belt, but calculated the P – T conditions of Meso- to Neoproterozoic crustal-derived TTG melts in the Eastern Block of the NCC. Accordingly, we reconstruct the late Neoproterozoic continental thermal structure of Dengfeng-Taihua terrane based on newly obtained crustal thickness, Moho temperature and classic geotherm models (Fig. 4; Chapman, 1986). The Moho geothermal gradient ($\sim 8\text{--}12$ °C/km) is essentially the slope of the geotherm line at the crust-mantle boundary, the basic relationship between

basal heat flow (q_B) and Moho geothermal gradient ($\frac{\Delta T}{\Delta x}$) is *Fourier's law*:

$$q_B = -k \frac{\Delta T}{\Delta x}$$

In this case, k is thermal conductivity ($2.6 \text{ W m}^{-1} \text{ K}^{-1}$). Above all, the definitions and indicative meaning of “apparent geothermal gradient” and “Moho geothermal gradient” are obviously different. The high apparent geothermal gradient ($>23 \text{ }^\circ\text{C/km}$) proposed by Huang et al. (2020) reflect the arc-forearc region, and the low Moho geothermal gradient ($\sim 8\text{--}12 \text{ }^\circ\text{C/km}$) in this study correspond to the low heat flow of the forearc (Currie and Hyndman, 2006). Consequently, we suggest that the southern margin of the Eastern Block of the NCC experienced accretionary to collisional orogenesis at a convergent plate margin in the late Neoproterozoic. See lines 242–244, please.

Finally, spending a few hours perfecting the English (which is fine) would make this even more readable and is probably worth the effort.

Revised. According to your suggestion, we have thoroughly polished the English language by the co-authors, and the technical expressions in the manuscript also have much improved.

Response to comments of Reviewer #2

When the plate tectonics launched on Earth is hereto hotly debated. This paper reports the Meso-Archean (ca. 3.0 Ga) switch of thermal state in the lower crust or Moho surface archived by the low Mg# TTGs in the Eastern block of North China Craton (NCC). The derivation results suggest the cooling of thermal gradients of Moho and thickening of crust resulted from hot subduction during the Meso- to Neo-Archean time. The authors state that this phenomenon probably typifies the transition of geodynamic regimes from divergent to convergent, as a function of modern-like plate tectonics (hot subduction). This contribution gives a very valuable insight into the evolving plate tectonics from the perspective of switch of the lithospheric thermal structure.

However, I still have concerns about the derivation procedure from the basic data to the

findings and therefore recommend a substantial major revision of this manuscript, after considering the reservations below, before publication.

(1) Generally, the value of Mg# typifies the degree to which the mantle proportion interacts and indicates the source of magma. It has been estimated that the bulk composition of the continental crust has Mg # of 34 to 42 (%). Here why did the authors select the value '45' as the dividing line between the magmas originating from the crust and the mantle (in Lines 54 to 60). In fact, this value was not, in particular, expressly stated in the cited papers as a magmatic source boundary (Rapp et al., 1999; Rapp and Watson, 1995; Stern and Kilian, 1996). On the contrary, Stern and Kilian (1996)'s work suggested when subducted oceanic basalt served as the source instead of the lower crust, the Mg# of the Austral Volcanic rocks (adakites) could be low to 40. Could the authors further explain the Mg# '45' option? If necessary, a plot or diagram illustrating the distribution of Mg# of all collected samples is better (more legible for readers) to set forth.

Revised. Thanks for your correction. After carefully checking, we have deleted the original text and revised the classification scheme for TTGs based on the SiO₂ versus Mg# diagram, and on this basis the TTGs can be clearly divided into two groups (Fig. 2c; Rapp et al., 1999; Rapp and Watson, 1995; Stern and Kilian, 1996). The low-Mg TTGs exhibit relatively low MgO contents and Mg# values, and plot in the compositional range of experimentally obtained partial melts of amphibolite and eclogite. Whereas the high-Mg TTGs display higher MgO contents and Mg# values than crustal partial melts obtained from experiment. See lines 112–120, please.

(2) What is the application condition on the use of Sr/Y and La/Y indicators for reconstructing crustal thickness? Based on the recent work (Figure 1a in Hu et al., 2020), the regression estimation of crustal thickness commonly produced larger errors due to data discretization in the thin crustal areas (~10-40 km), much less reliable than the thicker ones. How to judge or avoid the error when estimating thin crustal thickness by using the geochemical indicators, for instance when estimating the ~30-40 km Meso-

Archean crustal thickness in this contribution.

Revised. There are three basic application conditions for using Sr/Y and La/Yb ratios to reconstruct crustal thickness: (1) this method is constructed based on present oceanic subduction zone and continental collision zone, and exclude the application in intra-plate environment; (2) this method requires a large amount of data to ensure the reliability of the results; and (3) this method is based on intermediate-acid magmatic rocks (samples from subduction zones with SiO₂ of 55–70 wt.%, MgO of 1.0–6.0 wt.%, and Rb/Sr ratio of 0.05–0.20 and samples from collision zones with SiO₂ of 55–72 wt.%, MgO of 0.5–6.0 wt.% and average Rb/Sr ratio of < 0.35 were selected) (Chapman et al., 2015; Hu et al., 2017, 2020). The amount of TTG data in this study are relatively limited (a total of 155 samples) and tectonic environment is still uncertain. Moreover, the composition and thermodynamic state of continental crust are quite different between the Phanerozoic and Archean. Therefore, it is not appropriate to reconstruct the Archean crustal thickness by using the Sr/Y and La/Yb indicators. In the updated manuscript, we quantify the *P–T* conditions of the crustal-derived TTG melts by integrating the thermodynamic data with trace element simulations. Thermodynamic partial melting modelling calculates the mineral proportions of residual phases and melting degree under specific *P–T* conditions. We then simulate multiple trace element compositions of the partial melts based on the mineral proportion in the residual phases, melting degree and partition coefficient of trace elements using a simple batch partial melting model (Shaw, 1970), up to the chemical composition of modeled melts close to those of actual TTG samples, thus we could quantitatively calculate the pressure range of TTG melts. The pressure (*P*) conditions of crustal-derived TTG melts could represent the minimum estimate of crustal thickness, there are some variations in the estimation of crustal thickness (± 3.3 km, 2σ) due to the intrinsic errors in the pressure estimation (± 1 kbar, 2σ) (Palin et al., 2016b). See lines 134–150, please.

(3) In addition, the geochemical modeling only took the batch partial melting into account. Is the only paragenesis of the TTGs in NCC? How to exclude magma mixing

or assimilation and fractional crystallization (AFC) processes? Actually, some research progresses giving birth to new interpretations (models) that challenged the conventional consideration for the Mg# and generations of alleged ‘HP’ to ‘LP’ TTGs, which stressed melt segregation and differentiation of a single magma source in the continental crust (Getsinger et al., 2009; Laurent et al., 2020). How do the authors consider the paradox for the genesis of TTGs?

Revised. We agree with reviewer’s comment, and have revised the text. In fact, partial melting does not provide a unique petrogenesis of the Archean TTGs in the NCC (i.e., Bai et al., 2014; Smithies et al., 2019; Laurent et al., 2020). Previous investigations indicated that the Archean TTGs could be generated by partial melting of hydrated mafic rocks under different tectonic settings (including the over-thickened lower crust and subducted oceanic slab), or fractional crystallization of mantle-derived magmas. Given on the above paradox for the petrogenesis of TTGs, we have conducted a comprehensive petrogenetic classification of the TTGs in the updated text. Firstly, we filtered data by SiO₂ (>64 wt.%), Al₂O₃ (>13 wt.%) and Na₂O (3.0–7.0 wt.%) according to the TTG’s definition (Moyen and Martin, 2012), excluding non-typical TTG samples; Secondly, we removed TTG samples with heterogeneous whole-rock Nd and zircon Hf isotopic compositions, which may be derived from magma mixing or from contaminations of felsic crust as well as mantle peridotite during TTG magmatism; Thirdly, we determined which TTG samples that were derived from partial melting process through the petrogenetic discrimination; Finally, based on the experimental results of basalts, we removed the high-Mg TTG samples that may be affected by mantle materials. After carefully screening, only 155 samples of TTGs derived from melting of mafic rocks at the base of continental crust are used for estimating ancient crustal thickness and geothermal gradient in this study. See lines 50–70 and 97–122, please.

(4) A cooling trend at the lower continental crust from the Meso- to the Neo-Archean was unraveled in this contribution through numerations of basal heat flow and

metamorphic phase pseudosection modeling. From a practical perspective, was the temperature fall recorded by the rock or minerals in TTGs? Evaluating the crystallization temperatures of TTGs in the NCC via geo-thermometry, such as Ti-in-zircon and zircon saturation geothermometers, may provide straightforward mineralogic proof for the temperature drop or cooling trend.

Revised. Thanks for your suggestion. Thermodynamic and trace element modelling results show that the Moho temperature (magma temperature) gradually decreases from $\sim 850^\circ$, $\sim 810^\circ$ to $\sim 780^\circ \text{C}$ from the $\sim 2.9 \text{ Ga}$, $\sim 2.7 \text{ Ga}$ to $\sim 2.5 \text{ Ga}$ (Fig. 5b). This cooling trend at the lower continental crust is supported by mineralogical evidence. Titanium-in-zircon temperatures from the TTGs also were calculated using the equation by Ferry and Watson (2007): $\text{Log}(\text{ppm Ti-in-zircon}) = (5.711 \pm 0.072) - (4800 \pm 86)/T(\text{K}) - \text{Log} a(\text{SiO}_2) + \text{Log} a(\text{TiO}_2)$, where it is assumed that $\text{log} a(\text{SiO}_2) = 1$ because of the presence of quartz in the TTG and $\text{log} a(\text{TiO}_2) = 0.7$ from presence of titanite. Zircon may crystallize at different stages of magma evolution after the magmatic generation, rising and crystallization after emplacement, thus the temperature calculated from the Ti-in-zircon thermometer is generally lower than the temperature of magmatic generation (Corfu et al., 2003). Calculations reveal that zircon crystallization temperatures are $\sim 726^\circ$, $\sim 716^\circ$ and $\sim 682^\circ \text{C}$ for the $\sim 2.9 \text{ Ga}$, $\sim 2.7 \text{ Ga}$ to $\sim 2.5 \text{ Ga}$ TTGs, respectively, which is basically consistent with the decreasing trend of Moho temperature (Fig. 5b). See lines 167–168 and 576–577, please.

(5) The proportion of the residual phase basing the geochemical modeling were given to be some certain values (in Lines 127-133). However, the plots in Figures 2d-2f show ranges (or errors) of the residual proportion of garnet, instead of certain values, at each sampling locality. Do the ranges (or errors) affect much the pseudosection modeling result or not? Perhaps, this is the reason for the inconsistency between the thermodynamic modeling and the basal heat flux calculation. For instance, the thermal condition generating the $\sim 2.7 \text{ Ga}$ Zhongtiao TTG (in Figures 5a & 5b) is shown as $\sim 16^\circ \text{C}/\text{km}$ and the temperature can be deduced to be $\sim 720^\circ \text{C}$ (depth*thermal gradient).

While this sample is plotted at the conditions of ~ 900 °C and ~ 20 °C/km in the pseudosection by the contouring Sr/Y and garnet proportion (Figure 3). Furthermore, why were only 3 to 6 samples selected for plotting in pseudosection (Figure 3) and estimating thickness and thermal gradient (Figure 5)?

Revised. Thanks for your comments. Based on the issues that you indicated here, we have deleted the original Fig. 2d–f and relevant descriptions in the updated text, and have improved the “ P – T estimation method for TTG melts” as follows, 1) conduct a systematic classification of the Meso- to Neoproterozoic TTGs from the Eastern Block of the NCC, and select the purely crustal-derived TTG melts; 2) select the average composition of ~ 2.9 Ga, ~ 2.7 Ga and ~ 2.5 Ga low-K tholeiitic rocks from the Eastern Block of the NCC as the potential source of contemporaneous TTG melts, respectively; 3) carry out thermodynamic and trace element modelling using the potential source rocks and TTG melts; 4) quantitatively calculate the P – T conditions of TTG melts by comparing the chemical composition of modeled melts and actual TTG melts (Fig. 3). Based on the newly obtained melting P – T conditions, we reconstructed the geotherm model of early continental lithosphere (Fig. 4), and recalculated the Meso- to Neoproterozoic thermal state (including crustal thickness, basal heat flow and Moho geothermal gradient) of the Eastern Block, NCC (Fig. 5). It is noteworthy that the Moho geothermal gradient is essentially the slope of the geotherm line at the crust-mantle boundary, instead of the Moho temperature divided by Moho depth. See lines 97–122, 134–150 and 284–292, please.

Specific comments:

(1) Line 49, ‘constituting’ should be corrected as ‘constitutes’.

Revised. See line 47, please.

(2) Line 65, ‘quantitative’ should be corrected as ‘quantitatively’.

Revised. See line 75, please.

(3) Line 227, necessary and representative citation on the orogenic cycle of subduction in the Eastern Block during the early to late Neoproterozoic should be inserted behind.

Revised. See lines 235–247, please.

(4) Line 461 in the bibliography, Chapman (1986)'s paper was published in the <Geological Society, London, Special Publications>.

Revised. See lines 439–440, please.

Response to comments of Reviewer #3 (Prof. Richard Palin)

This manuscript details changing crustal (lithospheric) thickness across the North China Craton over a period of c. 2.9 Ga to c. 2.5 Ga, and the authors then examines whether these reported changes could have been driven by a transition from stagnant lid to mobile lid tectonics, as this time period is noted by many people as being a strong candidate for the timing of onset of global subduction. It does so by using a two-pronged approach. First, by analyzing the geochemistry and age of crustal-derived TTG gneiss from across the craton, with the idea that the maximum pressure of magma generation can constrain the maximum thickness of the continental crust at this time. Then, these pressures are used to build a geothermal model that varies in space and time. I have no issue with the methods and results of this second stage, but I have moderate to major concerns about the validity of assumptions and interpretations made as they relate to this first stage (i.e. working out “pressure” of formation of the TTG parent magma). One critically flawed assumption is that the same generic protolith can be used to calculate the garnet content of the residue from which these TTG magmas were derived. Many studies have shown that the metamorphic parageneses in the residue (and so the volume of melt and its composition generated) are highly sensitive to protolith composition. Further, I see no convincing evidence that these TTG have to have been derived at the base of over-thickened crust. The geochemical arguments for this are weak (and non-specific), and some of the sample set could equally have been derived via shallow subduction, which would give the impression of a thick overlying crust, where one did not exist. Finally, the “matching” of trace element ratios against interpreted garnet vol. % in the residue is shaky at best, and unfortunately getting robust values for pressure of magma generation is a cornerstone of the subsequent thermal

modelling. In principle, this approach is innovative and should lead to a high-impact study that would be of much interest to many workers worldwide studying Archean geodynamics. However, there are too many points of contention for me to recommend publication in its current form. I hope the authors find these comments useful.

Revised. Thanks for your comments. We have eliminated the confusing assumptions and simplifications, and the main changes can be emphasized as follows:

- 1) We have conducted a systematic petrogenetic classification of TTGs in the Eastern Block of the NCC. Firstly, a total of 287 TTG samples that align with the definition of Archean TTGs form a data subset of the TTGs (Fig. 2a; Moyen and Martin, 2012). Secondly, we ruled out the TTG samples with heterogeneous whole-rock Nd and zircon Hf isotopic compositions, which may be derived from magma mixing or from contamination of felsic crust as well as mantle peridotite during TTG magmatism. Thirdly, we applied petrogenetic discrimination using the highly (C^H) and moderately (C^M) incompatible element ratio (C^H/C^M) versus (C^H) diagram. Combining with the isotopic features above, most TTG samples distribute along a straight line that suggest a partial melting formation process. Samples showing a horizontal trend or a discrete distribution have been excluded because they may experience either magma mixing or assimilation and fractional crystallization (AFC) processes (Fig. 2b; Schiano et al., 2010). Finally, we removed the high-Mg TTG samples that affected by interaction with the ultramafic mantle peridotite based on previous experimental results of metabasalts (Fig. 2c; Rapp et al., 1999; Rapp and Watson, 1995; Stern and Kilian, 1996). After carefully screening, only 155 samples of TTGs were most likely derived at the base of thickened crust, which can be used for estimating ancient crustal thickness and geothermal gradient in this study. See lines 97–122, please.
- 2) We have changed the potential source rocks of the TTGs from the originally single source rock (global average composition of Archean enriched tholeiites) into three separated source rocks (average composition of ~2.9 Ga, ~2.7 Ga and ~2.5 Ga low-K tholeiitic rocks in the study area). See lines 284–292, please.

3) We have carried out thermodynamic and trace element modelling based on the updated potential source rock composition, `Perple_X` software and thermodynamic database. The mineral proportions of residual phases and melting degrees were obtained under specific P – T conditions by thermodynamic calculations of partial melting for specific sourced rocks, then, we simulated multiple trace element compositions of the partial melts based on the mineral proportion in the residual phases, melting degree and partition coefficient of trace elements using a simple batch partial melting model (Shaw, 1970), up to the chemical composition of modeled melts close to those of actual TTG samples, thus we could quantitatively calculate the P – T range of TTG melts. The pressure (P) conditions of crustal-derived TTG melts may be used to estimate the minimum crustal thickness, and the temperature (T) conditions representing the lower limit of the Moho temperature. Accordingly, we reconstructed the geotherm model of early continental lithosphere (Fig. 4), and recalculated the Meso- to Neoproterozoic thermal state (including crustal thickness, basal heat flow and Moho geothermal gradient) of the Eastern Block, NCC (Fig. 5). See lines 134–150, 293–306 and 314–316, please.

Line 37: Continental or oceanic lithosphere, or both? I'd like to see the writing tightened up considerably throughout to remove ambiguity such as this.

Revised. We have clarified it clear as the continental lithosphere. See line 37, please.

Lines 40–42: Please clarify (and/or simplify) this for the general reader. What exactly do you mean by “transition between different thermal states”? A decrease in mantle T_P ? Conduction vs. convection as the dominant mode of heat transport?

Revised. According to your advice, we have changed the original confusing expression into “the secular cooling of Earth’s mantle”. See lines 42–43, please.

Line 46: Missing word (or letter?) at the end of this sentence.

Revised. We have added “regimes” after “Archean geodynamic”. See line 46, please.

Line 47: Is ‘basement’ the right word here? Greenstones are not ‘basement’ lithologies. If you go to northwest Scotland, for example, you find high-grade amphibolites and mafic granulites, which form the true basement of Archean terranes.

Revised. We have changed the original sentence “Archean cratons are dominated by metamorphic basement terranes are composed of high-grade tonalite–trondhjemite–granodiorite (TTG) gneisses and granite-greenstone belts” into “Tonalite–trondhjemite–granodiorite (TTG) gneisses constitute a dominant part of all the granite-greenstone belts and high-grade terranes in globally preserved Archean cratons”. See lines 47–49, please.

Lines 47–49: Where does this value of 70 vol. % come from? The cited Martin et al. (2005) paper? More recent studies suggest that the Archean crust was more mafic than previously thought (e.g. Tang et al., 2016), with TTG gneiss making up subsidiary amounts, although that study does explicitly focus on the uppermost crust.

- Tang, M., Chen, K. and Rudnick, R.L., 2016. Archean upper crust transition from mafic to felsic marks the onset of plate tectonics. *Science*, 351(6271), pp.372-375.

Revised. The value of 70 vol. % in the original text is based on the research of Martin et al. (2005) and Moyen (2011). As reviewer’s said, the proportion of TTGs in the Archean continental crust is controversial. For instance, Martin et al. (2005) demonstrate that about 90% of the juvenile continental crust generated between 4.0 and 2.5 Ga belongs to ‘TTG suites’. Moyen (2011) consider that the Archean TTGs occupied about half the volume of continental crust. Whereas Tang et al. (2016) propose that the proportion of TTGs increased significantly from early Archean (4.0–3.0 Ga, 10–40%) to late Archean (3.0–2.5 Ga, >80%). Therefore, we have changed the “constituting some 70 % of the crustal volume” into “constitute a dominant part of all the granite-greenstone belts and high-grade terranes in globally preserved Archean cratons” in the updated manuscript. See lines 47–49, please.

Lines 54–55: This is a mistake, I assume? A slab melting model involves melting of a subducted slab (i.e. the basaltic portion of subducted oceanic lithosphere) – not devolatilization of subducted crust, infiltration of this fluid into the mantle wedge, and melting thereof. None of the papers cited here (refs. 24 to 26) suggest that fluid fluxing into the mantle creates TTG magmas. In any case, there are two distinct end-members of the slab melting hypothesis: shallow and steep (figure below from Palin et al., 2016): Shallow slab melting becomes very important for your discussion/sample selection later, because the lack of an intermediate mantle wedge makes TTG melts derived in those scenarios resemble melts formed at the base of a crustal column (i.e. they are not affected as much by interaction with the ultramafic mantle during ascent). I’d advise rewording this discussion of “end members” appropriately.

- Palin, R.M., White, R.W. and Green, E.C.R., 2016a. Partial melting of metabasic rocks and the generation of tonalitic–trondhjemitic–granodioritic (TTG) crust in the Archaean: Constraints from phase equilibrium modelling. *Precambrian Research*, 287, pp.73-90.

This is now revised. I think the situation of ‘partial melting of fluid fluxed mantle’ is a bit more complicated, and Bouilhol et al. (2015) addressed this in a paper (Bouilhol, P., Magni, V., Van Hunen, J., Kaislaniemi, L., 2015. A numerical approach to melting in warm subduction zones. *Earth and Planetary Science Letters*, 411, 37–44). During subduction in a hot regime (young, modern subduction zone, but also subduction in a hotter Earth), the crust dehydrates before it can melt. The dry crust then won’t melt anymore because its melting temperature is now too high (even probably in the Archaean). But the mantle lithosphere of the slab dehydrates later/deeper, and this fluid fluxes through the slab crust above, and this fluxed fluid can then lead to slab melting. According to your suggestion, we thoroughly revise the petrogenetic classification of Archean TTGs as following: (1) fractional crystallization of mafic melts from an enriched lithospheric mantle (Bai et al., 2014; Smithies et al., 2019; Laurent et al., 2020); (2) partial melting of hydrated mafic rocks at the base of continental crust (Moyen and Martin, 2012; Johnson et al., 2017; Fig. 1a); and, (3) melting of subducted oceanic slabs

(including steep and shallow subduction) (Palin et al., 2016a; Fig. 1b, c). Considering the complexity of the sources and formation processes of Archean TTGs, it is necessary to conduct a petrogenetic classification of TTGs in the NCC before discussion, only the TTG melts formed at the base of continental crust can be used for estimating the thermal state of early Earth. See lines 50–56, please.

Line 62, 66, etc: It's absolutely critical to tighten up the nomenclature here. The TTG gneisses are not the same as the TTG magmas generated from melting of a meta-basalt. The latter comes first and is later transformed into the former. Everywhere that you refer to TTG gneiss (incorrectly) you need to refer to the melt instead.

Revised. Thanks a lot, you are right. According to your suggestion, we have tightened up the nomenclature of the TTGs in the updated text. Namely, we changed “TTG gneiss” to “TTG melt”. See all the updated text.

Line 90–91: I am highly dubious about this statement and division. Given that only the low-MgO data are presented in supplementary information, it's not easy for me to check the rigor of this sample selection. Further, it seems from Fig. 1 that there is considerable overlap in geochemistry and mineralogy between both groups. It's evident that symbols for each overlap in broad regions. A heat map or adding some kind of transparency would be much more helpful to allow the reader to see how distinct they are in the region of commonality. The absolute values of MgO, for example (lines 98 and 102) do not seem sufficient to distinguish the two.

Revised. Thanks for your advice. After carefully checking, we deleted the original confusing expression and improved the classification scheme for TTGs. In the SiO₂ versus Mg# diagram, the TTGs can be clearly divided into two groups based on their distribution characteristics (Fig. 2c; Rapp et al., 1999; Rapp and Watson, 1995; Stern and Kilian, 1996). The low-Mg TTGs exhibit relatively low MgO contents and Mg# values, and plot in the compositional range of experimentally obtained partial melts of amphibolite and eclogite. Whereas the high-Mg TTGs display higher MgO contents

and Mg# values than crustal partial melts obtained from experiment. See lines 112–120, please.

I understand that there is a desire to filter the dataset to only include magmas generated via vertical burial and crustal thickening, but I think the authors need to do a much better job of eliminating the possibility that these melts could have formed in a shallow subduction zone (see figure above), which phase equilibrium modelling has shown is a viable geodynamic environment to form TTGs of appropriate mineralogy and chemistry matching natural Archean terranes (i.e. Palin et al., 2016a, and many others). Revised. Thanks a lot. It is difficult to effectively distinguish shallow slab-derived melts from thickened mafic crustal-derived melts according to the values of MgO and Mg# because both of them have not been affected by interaction with mantle peridotite during ascent. However, the shallow subducted oceanic slabs underplating along the crust-mantle boundary and eventually incorporated into the continental crust. Therefore, we considered that the melting P – T conditions of hydrated metabasalt in shallow subduction zone may record the thermal states of continental crust, too. See lines 54–56, 71–72 and 101–122, please. Based on the study of van Hunen et al. (2004) (van Hunen, J., van den Berg, A., Valaer, N.J., 2004. Various mechanisms to induce present-day shallow flat subduction and implications for the younger Earth: A numerical parameter study. *Physics of The Earth and Planetary Interiors* 146, 179–194), the global shallow-angle subduction is geodynamically viable, because if so, then oceanic plates were probably buoyant to cause the low-angle slabs, and if all oceanic lithosphere was buoyant, that would remove slab pull and would lead to no subduction at all, rather than shallow-angle subduction. Modern shallow-angle subduction, like under South America, is a local phenomenon, and can only exist because the majority of the Nazca plate is still negatively buoyant and drives the subduction process.

Line 118: Some data fall in the eclogite field of these discrimination diagrams? This gives more support for a shallow subduction environment, at least for some of your data.

You would need exceptionally thick mafic crust to generate an eclogite base (instead it's more likely to be mafic granulite).

Thanks for your suggestion. We understand that the low-Mg TTGs plot into the melt range of amphibolite and eclogite in the Mg# versus SiO₂ diagram (Fig. 2c), indicates the presence of garnet in the residual phases, and the TTG melts may be derived from either eclogites or amphibolites under relatively high *P–T* conditions. Furthermore, our estimation of *P–T* conditions also reveal that most TTG melts in the Eastern Block of the NCC distributed in a range from amphibolite facies (AM) to granulite (GN) and high-pressure granulite (HPG) facies (see following diagram), but under the univariant line of 'Ab = Jd + Qtz'. Therefore, the crustal-derived TTG melts in this study were mostly likely derived from garnet amphibolites or high-pressure granulites at the base of continental crust.

Lines 121–123: Arc basalts/tholeiites are extrusive rocks – is there a thought as to how they can end up as the juvenile source rocks that melt at lower-crustal pressures (depths) to form these voluminous TTGs?

Revised. A good question. Although the chemical composition of “basalts/tholeiites” and “basaltic/tholeiitic rocks” are very close, whereas the former refer to volcanic rocks that erupted to the surface, and the latter are mafic rocks may be emplaced by the underplating of mantle-derived magmas into the bottom and other levels of continental crust. The Archean low-K tholeiitic rocks may be the most appropriate source composition for the crustal-derived TTG melts in the Eastern Block of the NCC on the basis of their petrogenesis. Therefore, the “basalts/tholeiites” were changed into “basaltic/tholeiitic rocks” in the updated text. See lines 121, 136 and 284–292, please.

Lines 126–133: The line-fitting precision for these supposed volumes of garnet in the residue is extremely poor. The examples given for East Hebei and Anhui are both based on two points, none of which lie on the 10% garnet curve. The example for Jiaodong seems like it would fit considerably better with 25% or 30% garnet, not 20%. It would be easier to see these data if all lines with 5% increments were shown on each diagram. Was there any analytical calculations performed here to determine garnet volume, or was it simply curve fitting? Aside from this poor fit being a source of uncertainty, this analysis relies entirely on the assumption that the generic arc basalt used as the protolith is (a) correct and (b) chemically homogenous for all examples. Is it believable that the entire base of the NCC can be described by a single, chemically homogenous source rock?

Revised. Thanks a lot. Based on your advice, we have deleted the original Fig. 2d–f and relevant descriptions in the updated text. To solve this problem, we have modified the “ P – T estimation method for TTG melts” as follows, 1) conduct a systematic classification of the Meso- to Neoproterozoic TTG melts from the Eastern Block of the NCC, and select the crustal-derived TTG melts; 2) select the average composition of ~2.9 Ga, ~2.7 Ga and ~2.5 Ga low-K tholeiitic rocks from the Eastern Block of the NCC as the potential source of contemporaneous TTG melts, respectively; 3) carry out thermodynamic and trace element modelling using the latest determined mafic source rocks and TTG melts; 4) quantitatively calculate the P – T conditions of TTG melts by

comparing the chemical composition of modeled melts and actual TTG melts (Fig. 3). Based on the newly obtained melting P – T conditions, we reconstructed the geotherm model of early continental lithosphere (Fig. 4), and recalculated the Meso- to Neoproterozoic thermal state (including crustal thickness, basal heat flow and Moho geothermal gradient) of the Eastern Block, NCC (Fig. 5). See lines 97–122, 134–150 and 284–292, please.

Lines 133–136: This is just too broad of a statement/assumption. There are several variables that could account for this – not solely pressure – that are not justified. Many studies using experimental petrology and/or phase diagram analysis show that garnet stability is also temperature and bulk composition dependent. Presumably you can see this for yourself in the diagram calculated for Figure 3?

Revised. Thanks for your comment. We have modified the “ P – T estimation method for TTG melts”, and the original statement has been deleted in the updated version. In the revised version, the potential source rocks of the TTGs was changed from the originally single source rock (global average composition of Archean enriched tholeiites) into three separated source rocks (~2.9 Ga, ~2.7 Ga and ~2.5 Ga low-K tholeiitic rocks in the study area). The mineral proportions of residual phases were obtained under specific P – T conditions by thermodynamic calculations of partial melting for specific sourced rocks, and the blue dashed contours in the P – T phase diagrams represent the calculated garnet proportion (%) in residue (Fig. 3). Then, we simulated multiple trace element compositions of the partial melts based on the mineral proportion in the residual phases, melting degree and partition coefficient of trace elements using a simple batch partial melting model (Shaw, 1970), up to the chemical composition of modeled melts close to those of actual TTG samples, thus we could quantitatively calculate the pressure range of TTG melts, namely, the ancient crustal thickness. See lines 134–150 and revised Figure 3, please.

Line 139: I see in the Methods that you used a CIPW norm assessment to determine

this water content. If you're using the bulk composition of Archean arc-like basalt from Table S5 for the rest of the modelling, why not just use the (measured) average LOI? Isn't it more reliable than introducing unnecessary assumptions when dealing with the normative anhydrous results?

Revised. Thanks for your suggestion, we have deleted the "Determination of source water content [X(H₂O)]" in the Methods part. The estimation for the water content have been changed to be calculated directly from the average LOI of the potential mafic source rocks. Accordingly, the water contents of ~2.9 Ga, ~2.7 Ga and ~2.5 Ga low-K tholeiitic rocks were 1.2, 1.8 and 2.0 wt.%, respectively. See lines 136–137, please.

Lines 140–142: This short sentence touched on a conceptual issue that I raised before. The TTG magmas are not the same as the TTG gneisses. How do you take into account the magmas separating from their source, ascending through the crust, stalling, and crystallizing? Is this change in pressure significant? Could trace element ratios have been changed during magma ascent, thus calling into question your prior assumptions of garnet proportion in the residue?

Revised. According to your advice, we carry out a systematic genetic classification of Archean TTGs before discussing the thermal state of continental crust. Firstly, we filtered data by SiO₂ (>64 wt.%), Al₂O₃ (>13 wt.%) and Na₂O (3.0–7.0 wt.%) according to the TTG's definition (Moyen and Martin, 2012), excluding non-typical TTG samples; Secondly, we removed TTG samples with heterogeneous whole-rock Nd and zircon Hf isotopic compositions, which may be derived from magma mixing or from contaminations of felsic crust as well as mantle peridotite during TTG magmatism; Thirdly, we determined which TTG samples that were derived from partial melting process through the petrogenetic discrimination; Finally, based on the experimental results of basalts, we removed the high-Mg TTG samples that may be affected by mantle materials. Our selected crustal-derived TTGs display low loss on ignition (LOI) values of less than 2.0 wt.%, and lack of significant Ce anomalies ($\delta Ce = Ce_N / \sqrt{La_N \times Pr_N}$) with values of 0.79–1.31, indicating that the rocks have not been subjected to

any major alteration (Supplementary Table 4; Polat and Hofmann, 2003). Therefore, the chemical composition of the crustal-derived TTGs is considered to be close to that of the TTG magmas. Therefore, we changed “TTG gneiss” to “TTG melt” in the updated text. See lines 97–122 and 278–283, please.

Lines 143–145: Given the many intrinsic sources of uncertainty in phase diagram modelling (e.g. Powell and Holland, 2008), it’s really not appropriate to report calculated linear geotherms with a precision of one decimal place. Just simplify them (e.g. 18.2–22.8 °C/km -> 18-23 °C/km). To be completely rigorous, you probably should take into account the standard uncertainty on P-T estimation using phase equilibrium modelling (± 50 °C and ± 1 kbar at 2 sigma: Palin et al., 2016b).

- Powell, R. and Holland, T.J.B., 2008. On thermobarometry. *Journal of Metamorphic Geology*, 26(2), pp.155-179.
- Palin, R.M., Weller, O.M., Waters, D.J. and Dyck, B., 2016. Quantifying geological uncertainty in metamorphic phase equilibria modelling; a Monte Carlo assessment and implications for tectonic interpretations. *Geoscience Frontiers*, 7(4), pp.591-607.

Revised. According to your advice, we keep the integers when reporting the calculated linear geotherms (Table 1). Furthermore, the standard uncertainty on $P-T$ estimation was modified to ± 50 °C and ± 1 kbar at the 2 sigma level (Palin et al., 2016b). See lines 145–146, please.

Lines 158–159: What are these thicknesses when you take into account uncertainty in pressure estimation (cf above)? What was your pressure-to-depth conversion ratio? Was it constant throughout the crust? If so, why, as the overburden likely changes with absolute depth (i.e. upper crust is more volcanic and potentially mafic greenstone, so has a different density to the felsic pluton-rich middle crust)? You are hanging your hats on these changes in depth, so the reader should be fully briefed on the uncertainties on the numbers and how they were derived.

Revised. Assuming that the tectonic overpressure for the continental crust is negligible

and that 1 GPa \approx 33 km crustal depth, the pressure (P) range of crustal-derived TTG melts can be used to estimate the lower limit of ancient crustal thickness. Moreover, there are some variations in the estimation of crustal thickness (± 3.3 km, 2σ) due to the intrinsic errors in the pressure estimation (± 1 kbar, 2σ). See lines 146–150, please.

Line 189: This is not “relatively thin” compared to modern-day average continental crust and in my mind further supports the possibility that some TTG magmas were generated at depths exceeding that of the crust (i.e. in a shallowly subducted slab). Have you considered how these numbers weigh up against independent estimates of Archean continental crust thicknesses (e.g. Dhuime et al., 2015)?

- Dhuime, B., Wuestefeld, A. and Hawkesworth, C.J., 2015. Emergence of modern continental crust about 3 billion years ago. *Nature Geoscience*, 8(7), pp.552-555.

Thank you for your comment. First of all, the “relative thin” here is compared to the Neoproterozoic (~ 2.7 – 2.5 Ga) crustal thickness, but not to that of the modern continental crust. Second, in the petrogenetic classification of TTGs mentioned above, we have removed the TTG samples affected by mantle peridotites, only the crustal-derived TTG melts were used to estimate the paleo-crustal thickness. Dhuime et al. (2015) calculate the global-scale variation in the thickness of juvenile continental crust based on the Rb/Sr ratios of the juvenile continental crust, whereas our study estimate the Meso- to Neoproterozoic crustal thickness at the craton-scale (NCC) by combining the thermodynamic and trace element modelling results of crustal-derived TTG melts. Importantly, Dhuime et al. (2015) did not calculate thickness at time of melting to produce TTG’s but rather at time the crust was extracted from the mantle. Above all, these two estimation methods are different in terms of research object and research scope, thus it is reasonable that there is some difference in the estimated crustal thickness.

Lines 209–217: This is the first mention I can see of possible shallow subduction. This must be discussed much earlier in the manuscript.

Revised. We have added the shallow subduction model for Archean TTG magma genesis into the Introduction part. See lines 53–56 and 71–72, please.

Lines 255–256: I'm sorry to keep bringing this up, but subsequent studies have shown that this tripartite classification simply leads to confusion. The figure on the next page (again from Palin et al., 2016a) shows mineral proportions in the residue (specifically garnet and plagioclase) and melt fraction generated from an enriched Archean tholeiite – like your example – at a range of P-T conditions appropriate for TTG generation. Garnet proportions of >30 vol. % can be generated at only moderate pressures (~12-14 kbar = 36-42 km depth) at only moderate temperatures (~850-900 °C). All of Moyen's work is of exceptional quality, but the advent of petrological modelling and forward calculation of trace element ratios and residual mineral assemblages can demonstrate easily that high-pressures (>20 kbar, according to the early studies, e.g. Rapp et al., 1990) are not required to generate a TTG with a “high-pressure” signature. This is just a fact, and I assume that you accept the value and reliability of this kind of thermodynamic modelling, as you employ it in your own study here.

Revised. Thank you for your suggestion, we have removed the “Implications for global-scale ancient cratons” from the discussion section.

Methods

Line 319: What rock? Your bulk composition is a statistical average, isn't it? How do you know the source rocks don't contain apatite? If they don't, where does the P₂O₅ reside?

Revised. To guarantee the estimation results more reliable, we change the potential source rock composition from “enriched Archean tholeiites worldwide” to “Meso- to Neoproterozoic low-K tholeiitic rocks in the Eastern Block of the NCC”. In the past decade, our research group carried out detailed investigations of geology, lithological assemblages and petrogenesis for Archean supracrustal rocks in the NCC. Because the mode of apatite and the bulk content of P₂O₅ are very low in the studied rocks, and P₂O₅

is currently not considered as a component in the thermodynamic database, we choose a system Na₂O-CaO-K₂O-FeO-MgO-Al₂O₃-SiO₂-H₂O-TiO₂-O₂ (NCKFMASHTO) for the thermodynamic modelling. See lines 297–301, please.

Lines 323–327: This must be an error, because the Green et al. (2016) a-x relations are not compatible with the pre-2011 Holland and Powell dataset. Can you confirm this?

Revised. After the replacement of mafic source rock, we conduct the thermodynamic modelling of Meso- to Neoproterozoic low-K tholeiitic rocks in the Eastern Block of the NCC. Latest calculations were performed with updated Perple_X software (version 6.9.0) in the Na₂O-CaO-K₂O-FeO-MgO-Al₂O₃-SiO₂-H₂O-TiO₂-O₂ (NCKFMASHTO) system using the hp633ver data set, assuming $Fe^{3+}/(Fe^{3+} + Fe^{2+}) = 0.1$ (Connolly and Kerrick, 2002; Holland and Powell, 2011; Ge et al., 2018). Recently released solution models are used for the melt, amphibole and clinopyroxene (Green et al., 2016); garnet, orthopyroxene, ilmenite and mica (White et al., 2014); plagioclase (Holland and Powell, 2003) and biotite (Powell and Holland, 1999). Quartz, rutile, titanite, zoisite, magnetite and water are considered pure end-members. See lines 293–306, please.

Lines 560–563/Fig. 3: The contours (dashed lines) for garnet % and Sr/Y in the melt cannot be correct, and I am always hesitant to make such statements for phase diagrams that I have not produced, but there are clear problems with them. I have enough experience producing these pseudosections to know that the slopes of isopleths that are controlled by intensive variables (i.e. mass of a phase) change across field boundaries. See the garnet volume contours on the figure from Palin et al. (2016a), next page, as an example. For instance, one of the garnet contours marked here for 15 vol. % intersects the garnet-in field boundary. This cannot be true – there cannot be simultaneously 10 vol. % and 0 vol. % of garnet at the same P-T condition (~0.9 GPa and ~910 °C). Secondly, how were the Sr/Y ratios calculated? Why don't they show similar trends to the garnet proportions as you are in a closed (bulk composition constrained) system? On that note, why is melt loss not considered here? It is thought that melt escapes in

pulses once a critical threshold is reached. There are too many small issues with this phase diagram for me to be confident that the calculated data (i.e. pressures) are reliable. Revised. Thank you for your valuable advice, with reference to the P – T diagrams in your article, we thoroughly modified the “thermodynamic and trace element modelling”. After carefully revising, we got three new P – T phase diagrams based on the updated source rock composition, Perple_X software and thermodynamic database (Fig. 3). The equilibrium mineral assemblages were calculated at discrete P – T points for every 10°C and 0.1 GPa from 750° to 950°C and 0.6 to 2.0 GPa, with a water contents ranging from 1.2 to 2.0 wt.% (Supplementary Table 7). The black and blue dashed contours in the P – T phase diagrams represent the calculated degree of melting (wt.% of melt) and garnet proportion (%) in residue, respectively (Fig. 3a, c and e). Finally, we can quantitatively calculate the P – T ranges of crustal-derived TTG melts with different ages by combining the thermodynamic and trace element modelling results. See lines 293–306, 314–316 and 557–565, please.

Supplementary Information

Fig S11 – What are the grey ellipses on these plots? Please expand the caption.

Revised. The grey ellipses on the Fig S11 have no specific meaning, thus we have removed them. See the updated Fig S11, please.

Selection of mafic source rock – this result (“Geochemical modeling results clearly show that most MORBs and OPBs are too depleted in LILEs to generate Archean TTGs, whereas the IABs are slightly too enrich. The most likely mafic source rock for Archean TTGs is Archean ‘arc-like’ basalt”) has been rigorously demonstrated previously by Martin et al. (2014). Why was this treatment repeated here?

- Martin, H., Moyen, J.F., Guitreau, M., Blichert-Toft, J. and Le Pennec, J.L., 2014. Why Archean TTG cannot be generated by MORB melting in subduction zones. *Lithos*, 198, pp.1-13.

Revised. There is consensus that the crustal-derived TTGs were derived from

dehydration melting of metabasalts, and extensive experimental and petrological studies show that the enriched Archean tholeiites are ideal protolith for TTGs (i.e., Martin et al., 2014). Consequently, we have deleted all the unnecessary repetition according to your advice. See the updated Supplementary Information, please.

Reviewers' Comments:

Reviewer #1:

Remarks to the Author:

The authors have done a decent job of addressing most of the points, although the responses are rather long-winded and repetitive. A few points to consider.

My comment "It's not entirely clear why 'tonalitic-trondhjemitic-granodioritic gneisses are ideal lithologies for reconstructing the thermal state of early continental crust'. From a phase equilibrium perspective, they generally contain high-variance assemblage stable over large P-T fields." Has not been addressed. The fact remains that TTGs are not ideal for the job, but they are pretty much all we have to work with.

I think the role of H₂O-fluxed melting could be mentioned when discussion compositional differences in TTGs.

A long-standing bugbear is in invoking 'plumes' as the driving force for melting and crust formation. This is mainly a semantic point, but to most people 'plumes' are generated from thermal anomalies at the core-mantle boundary. In this definition, I don't see any requirement for plumes in the Archean. Mantle upwellings (which the authors refer to) and/or thermal anomalies within the upper mantle can do the job required. In short, I'd either stick to mantle upwellings, which of course can include plumes *sensu stricto*, or describe more concisely what you mean when you use the term 'plume'.

Tim Johnson

Reviewer #2:

Remarks to the Author:

I have now read this revised manuscript and the detailed point-by-point response to the reviewer's comments. The authors have done a serious major modification and improvement on some key aspects of the method, thermodynamic calculation, sample filtration and interpretation based on their data. I am generally convinced that the authors have adequately addressed all the concerns and critiques. I recommend the paper be accepted and look forward to seeing this contribution published in Nature Communications.

Wenjiao Xiao

Reviewer #3:

Remarks to the Author:

Review of: "Thermal state and evolving geodynamic regimes of the Meso- to Neoproterozoic North China Craton"

The authors have done a good job addressing my initial comments and the manuscript is substantially revised and improved. One notable concern from the first version was the poor fitting of trace element data to (interpreted) garnet volume in the melt-bearing precursor rock, which formed a primary argument for a changing crustal thickness through time. This has been improved by applying thermobarometry and petrological modelling to determine calculated garnet proportions in specific precursor lithologies. This is a significant improvement on what I considered to be a potential weak link in the original study. I would, however, urge caution in considering unusually high-Fe Archean basalt from the Pilbara craton as a suitable precursor lithology for the TTG magmas. This modelling work was initially performed by Johnson et al. (2017), although after publication, a Corrigendum was

issued to alert readers that those basalts have a significantly lower Mg# to most Archean basalts of a similar age. This artificially expands the garnet stability field to lower pressure, and gives a false impression of the 'shallow levels of the crust' where some TTGs can be formed. These Pilbara basalts are strongly atypical in composition for their age and probably not applicable to the NCC. The Corrigendum states the following:

In this Letter we omitted to cite a paper (Palin et al., 2016) that also used recently developed thermodynamic models (Green et al., 2016) to predict the melting process in Archean metabasaltic rocks. Importantly, the average enriched Archean tholeiite used by Palin et al. (2016) as a proposed source rock (Condie, 1981) for tonalite-trondhjemite-granodiorite rocks has a magnesium number (Mg#) of 57, significantly higher than the average value for the CF-2 basalts (with Mg# of 35). This difference has profound implications for the results of these studies. We regret not citing Palin et al. (2016) to emphasize the clear distinction between their findings and those of our study. The original Letter has not been corrected.

- Johnson, T.E., Brown, M., Gardiner, N.J., Kirkland, C.L. and Smithies, R.H., 2017. Corrigendum: Earths first stable continents did not form by subduction. *Nature*, 545, 510-510.

Despite this, I don't expect that the authors would go through and revert these changes completely. I just think it worthwhile (a) being aware of this problem and/or (b) making note of this atypical bulk composition somewhere in the text.

The remainder of the changes that have been performed during revision are very good and the work is much clearer now.

Richard Palin

Response to referees letters

Response to comments of Reviewer #1 (Prof. Tim Johnson)

The authors have done a decent job of addressing most of the points, although the responses are rather long-winded and repetitive. A few points to consider.

My comment “It’s not entirely clear why ‘tonalitic–trondhjemitic–granodioritic gneisses are ideal lithologies for reconstructing the thermal state of early continental crust’. From a phase equilibrium perspective, they generally contain high-variance assemblage stable over large P – T fields.” Has not been addressed. The fact remains that TTGs are not ideal for the job, but they are pretty much all we have to work with.

We agree with the Prof. Johnson that there are complexities in TTG genesis. But as he also noted, TTGs are the most widespread magmatic differentiation products in the Archean and much of our understanding of the Archean crust comes from TTGs (e.g., Palin et al., 2016; Johnson et al., 2017; Ge et al., 2018; Wiemer et al., 2018; Smithies et al., 2019, 2021; Laurent et al., 2020). In this manuscript, we also stress that not all TTGs are suitable for reconstructing the thermal state of the early Earth’s continental crust given their potentially complex origins (lower crust melting, slab melting and fractionation of mafic melts).

Although different types of TTG may be formed under similar P – T conditions, their genesis may be distinguished according to their geochemical and isotopic characteristics. Therefore, we only choose the TTG samples that derived from partial melting of crustal mafic rocks as the focus for this study. Constraints from experimental petrology indicate that the TTGs derived from melting of mafic rocks are mainly formed at the root of the continental crust, which is close to the boundary between the crust and the lithospheric mantle (namely, the Moho surface) (Rapp and Watson, 1995; Qian and Hermann, 2013). In this way, the P – T conditions of the studied TTGs can be used to constrain the Meso- to Neoproterozoic thermal state of the Moho surface in the eastern North China Craton.

To better define the P – T conditions generating the TTGs from crustal mafic rocks, we have followed the following methodology: (1) select mafic rocks that have been

preserved in the Meso- to Neoproterozoic greenstone belts of the eastern NCC, and these mafic rocks show the similar zircon U–Pb ages, Hf isotope and whole-rock Nd isotopic features to those of the studied TTGs; (2) carry out the phase equilibrium simulation by using the composition of potential mafic source rocks and source water content (defined by the average LOI), and the mineral proportion of residual phases and melting degree were obtained under specific P – T conditions; (3) simulated multiple trace element compositions of the partial melts based on the mineral proportion in the residual phases, melting degree and partition coefficient of trace elements using a simple batch partial melting model (Shaw, 1970), up to the chemical composition of modeled melts close to those of actual TTG samples, thus we could quantitatively calculate the P – T range of TTG melts, and further calculate out the P – T means and errors in every studied area. In this way, the P – T conditions generating TTG melts in every region can be limited to a reasonable range, which are applied to constrain the smallest crust depth and the dT/dP as the closest geothermal gradient of the Moho surface in that region. See lines 76–82 and 145–169, please.

I think the role of H₂O-fluxed melting could be mentioned when discussion compositional differences in TTGs.

Revised. Thank you for the suggestion, and we have emphasized the importance of H₂O-fluxed melting in the ‘Introduction’ part. H₂O-fluxed melting plays a crucial role in the formation of slab-derived TTG melts (Palin et al., 2016), which contributes to the compositional diversity of the TTGs. Thermodynamic and trace element modelling results show that LILEs and LREEs of TTG melts are sensitive to water contents (Ge et al., 2018; Hu et al., 2021). According to the partial melting mechanism, the melting of mafic rocks at the bottom of continental crust is mainly controlled by the dehydration melting reaction of amphibole under the fluid-deficiency condition. Therefore, as suggested by another reviewer Richard Palin, we use the measured average LOI of the bulk composition of Archean tholeiitic rocks to determine the water content. See lines 56–58 and 74, please.

A long-standing bugbear is in invoking ‘plumes’ as the driving force for melting and crust formation. This is mainly a semantic point, but to most people ‘plumes’ are generated from thermal anomalies at the core–mantle boundary. In this definition, I don’t see any requirement for plumes in the Archean. Mantle upwellings (which the authors refer to) and/or thermal anomalies within the upper mantle can do the job required. In short, I’d either stick to mantle upwelling, which of course can include plumes *sensu stricto*, or describe more concisely what you mean when you use the term ‘plume’.

Revised. We agree with the reviewer. We have changed the “mantle plume” into “mantle upwelling” and/or “vertical tectonic regime” in the updated version of the manuscript. The development of komatiites, extension-related deformation in the Eastern Block of the North China Craton, and high-mantle potential temperature, indicated that vertical tectonic regime should play a dominant role in the crustal growth and evolution during the late Mesoproterozoic. See lines 30, 212, 214–215 and 218, please.

Tim Johnson

Response to comments of Reviewer #2 (Prof. Wenjiao Xiao)

I have now read this revised manuscript and the detailed point-by-point response to the reviewer’s comments. The authors have done a serious major modification and improvement on some key aspects of the method, thermodynamic calculation, sample filtration and interpretation based on their data. I am generally convinced that the authors have adequately addressed all the concerns and critiques. I recommend the paper be accepted and look forward to seeing this contribution published in Nature Communications.

Thank you very much for your kind comments.

Wenjiao Xiao

Response to comments of Reviewer #3 (Prof. Richard Palin)

The authors have done a good job addressing my initial comments and the manuscript is substantially revised and improved. One notable concern from the first version was the poor fitting of trace element data to (interpreted) garnet volume in the melt-bearing precursor rock, which formed a primary argument for a changing crustal thickness through time. This has been improved by applying thermobarometry and petrological modelling to determine calculated garnet proportions in specific precursor lithologies. This is a significant improvement on what I considered to be a potential weak link in the original study. I would, however, urge caution in considering unusually high-Fe Archean basalt from the Pilbara craton as a suitable precursor lithology for the TTG magmas. This modelling work was initially performed by Johnson et al. (2017), although after publication, a Corrigendum was issued to alert readers that those basalts have a significantly lower Mg# to most Archean basalts of a similar age. This artificially expands the garnet stability field to lower pressure, and gives a false impression of the 'shallow levels of the crust' where some TTGs can be formed. These Pilbara basalts are strongly atypical in composition for their age and probably not applicable to the NCC. The Corrigendum states the following:

In this Letter we omitted to cite a paper (Palin et al., 2016) that also used recently developed thermodynamic models (Green et al., 2016) to predict the melting process in Archaean metabasaltic rocks. Importantly, the average enriched Archaean tholeiite used by Palin et al. (2016) as a proposed source rock (Condie, 1981) for tonalite–trondhjemite–granodiorite rocks has a magnesium number (Mg#) of 57, significantly higher than the average value for the CF-2 basalts (with Mg# of 35). This difference has profound implications for the results of these studies. We regret not citing Palin et al. (2016) to emphasize the clear distinction between their findings and those of our study. The original Letter has not been corrected.

- Johnson, T.E., Brown, M., Gardiner, N.J., Kirkland, C.L. and Smithies, R.H., 2017. Corrigendum: Earths first stable continents did not form by subduction. *Nature*, 545,

510-510.

Despite this, I don't expect that the authors would go through and revert these changes completely. I just think it worthwhile (a) being aware of this problem and/or (b) making note of this atypical bulk composition somewhere in the text.

The remainder of the changes that have been performed during revision are very good and the work is much clearer now.

Revised. Thank you for the reminder. We did not use the Archean Pilbara basalts as the original component of the simulation. Based on another reviewer Tim Johnson's suggestion, we have introduced his work in the first round of revision. According to your suggestion, the Archean Pilbara basalt may not be a suitable source rock for the TTG magmas due to their atypical bulk rock composition. Therefore, we have added a statement of "However, these Pilbara metabasaltic source rocks have high TFeO contents resulting in their Mg# values being significantly lower than the average value of TTGs, which expands the garnet stability field to lower pressure (<1.0 GPa) (cf. Corrigendum to Johnson et al., 2017)" following "Johnson et al. (2017) demonstrated that the amount of garnet in residues is strongly dependent on the Mg# values ($Mg\# = Mg/(Mg + Total\ Fe^{2+}) \times 100$) of the metabasaltic source rocks, and proposed that MP TTGs may be stable at 0.7 GPa (~25 km)". See lines 68–71, please.

Richard Palin

Response to referees letters

In consideration of no comments by three reviewers (Prof. Tim Johnson, Prof. Wenjiao Xiao, Prof. Richard Palin) at this stage, we would like to appreciate their efforts in reviewing our manuscript at present. Their comments have led to significant improvement in the quality of the updated manuscript.